# Effects of combining feedback and hypothesis-testing on the quality of simulated child sexual abuse interviews with avatars among Chinese university students

Yiwen Zhang [1]*, Siyu Li[2,3], Yikang Zhang[1], Shumpei Haginoya[4], Pekka Olavi Santtila[3,5]*

1 Faculty of Psychology and Neuroscience, Maastricht University, Maastricht, The Netherlands, 2 The School of Psychology and Cognitive Science, East China Normal University, Shanghai, China, 3 Faculty of Arts and Sciences, New York University, Shanghai, China, 4 Faculty of Psychology Graduate School of Psychology, Meiji Gakuin University, Tokyo, Japan, 5 NYU-ECNU Institute for Social Development, Shanghai, China

* ophelia.y.zhang@outlook.com (YZ); pekka.santtila@nyu.edu (POS)

**Data Availability Statement:** All data and analysis code are available from OSF: https://doi.org/10.17605/OSF.IO/VFZH3.

## Abstract

Previous research has shown that simulation training using avatars with repeated feedback improves child sexual abuse interview quality. The present study added a hypothesis-testing intervention and examined if the combination of two interventions, feedback and hypothesis-testing, would improve interview quality compared to no intervention and to either intervention alone. Eighty-one Chinese university students were randomly assigned to a control, feedback, hypothesis-testing, or the combination of feedback and hypothesis-testing group and conducted five simulated child sexual abuse interviews online. Depending on the assigned group, feedback on the outcome of the cases and question types used in the interview were provided after each interview, and/or the participants built hypotheses based on preliminary case information before each interview. The combined interventions group and feedback group showed a higher proportion of recommended questions and correct details from the 3rd interview onward compared to the hypothesis-building and control groups. The difference between the number of correct conclusions was not significant. hypothesis-testing alone exacerbated the use of non-recommended questions over time. The results show that hypothesis-testing may impact question types used negatively but not when combined with feedback. The potential reasons for hypothesis-testing alone not being effective and the differences between the present and previous studies were discussed.

## Introduction

In 2020, China's Procuratorate prosecuted 15,365 persons for rape of minors, 5,880 for molesting children, and 1,461 persons for forcibly molesting and insulting minors, with increases of 19%, 15%, and 12%, respectively, compared with 2019 [1]. The seriousness and the increasing

**Funding:** Start-Up Grant from NYU Shanghai to author Pekka Santtila. The funders had no role in study design, data collection and analysis, decision to publish, or preparation of the manuscript.

**Competing interests:** The authors have declared that no competing interests exist.

numbers of child sexual abuse (CSA) cases underlines the urgency of establishing evidence-based training for conducting such investigations. Improving the quality of the investigative interviews is of central importance as in many CSA cases, the child's statement often serves as the most crucial piece of evidence, since there are rarely other pieces of evidence available in these cases [2–4].

Human memory is a reconstructive process subject to situational influences [5], making the quality of statements provided by both adults [6–8] and children [9,10] not merely a function of interviewee's memories but also of the interview process. Therefore, in investigative settings where the accuracy of memory reports is of great importance, interviewers should avoid exerting social influences that could contaminate the memories of the interviewees. This is especially true for child interviewing since children are, in general, more susceptible to suggestion than adults [11–14]. Therefore, adopting open-ended questions and avoiding closed-ended questions when interviewing children is an important recommendation [15–17]. In reality, however, the quality of investigative interviews with children remains a concern in various countries because of the continuing heavy use of closed questions, which may result in distortions of testimony, including contradictions and inaccurate details [18–21]. Given that the problem with the quality of investigative interviews is an issue worldwide and that the development of evidence-based training programs in China is still in its infancy, it is logical to expect that China also faces similar problems in CSA investigations.

In order to increase and maintain the use of open-ended questions in investigative interviews, different training programs have been developed and implemented. Nevertheless, there is a gap between knowing what is the best practice and the actual adoption of the appropriate question types during an interview. Programs that provide merely theoretical knowledge are not efficient in improving the interview quality [22–24]. Lamb et al. [25] showed that by giving a structured interview protocol to the interviewer, including the types of questions to use at different stages, combined with detailed feedback after interviews, their quality could be improved. However, this solution entails significant costs, and the training effects tend to disappear soon after feedback is not provided [25]. Besides, when real cases are used for training, the only type of feedback that can be provided to the interviewer concerns the questions used. No feedback can usually be given regarding whether the interviewer actually found out what had happened or not.

As a solution, a serious gaming training protocol that uses simulated CSA interviews with computer-generated avatars (hereafter, Avatar Training) has been proposed [26]. Avatar training, designed to promote the quality of investigative CSA interviews, creates a realistic interviewing setting for the interviewer to question a child avatar concerning suspected sexual abuse and gather information from the avatar as if they were real children. In this program, not only can interviewers receive feedback on the question types they used after each simulated interview, but also feedback on the outcome (i.e., whether the interviewer arrived at the correct conclusion). A series of experiments [27–30] has shown that simulated interviews combined with feedback on the outcomes and the question types resulted in more open-ended questions and less closed questions being used compared to the use of simulated interviews alone. Importantly, this training effect transferred into interviews with real children about a mock event sharing features with sexual abuse [31] as well as to actual investigative interviews in criminal cases [32]. Regarding the applicability of Avatar Training, previous studies conducted in both Western contexts [26,29,30,31] as well as in Japan [27,28,33], have supported its effectiveness.

The adoption of a less suggestive questioning style, however, does not adequately address another important issue in investigative interviews, which is, confirmation bias. Confirmation bias refers to a universal human tendency to look for and interpret evidence that is consistent

with one's prior beliefs [34,35]. Literature on information acquisition has clearly shown that people have a tendency to seek evidence confirming their initial belief [36–39], and to ignore information that goes against it [40,41], with potentially negative effects on professionals investigating CSA allegations [12,42–44]. Hence, pre-interview beliefs held by interviewers are central and may lead to confirmation bias, resulting in use of closed-ended questions [45,46] that may endanger the reliability of responses elicited from children and in extreme cases can make the interviewer arrive at an incorrect conclusion [47–49]. Besides, previous experiments using the child avatars have also found that a preliminary assumption of abuse resulted in more frequent use of non-recommended question types [35]. However, even if suggestive questions are absent, open questions may still focus on a particular assumption about what has happened. Focusing solely on question types may, therefore, be inadequate to address the issue of confirmation bias in CSA investigative interviews.

To address the problem with confirmation bias, we decided to investigate the impact of adding a hypothesis-building intervention to the Avatar Training. Formulating alternative hypotheses prior to interviewing a child has been suggested multiple times as a potential technique to counteract confirmation bias and to reduce false positive conclusions [42,50–52]. Hypothesis-building here refers to the practice where, after a careful assessment of background information but before the interview, interviewers formulate a series of alternative hypotheses about the case, with particular attention paid to how the allegation came about, risk factors for abuse, potential risks for pre-interview suggestive influences or misunderstandings [50]. Using a hypothesis-testing approach is considered central in avoiding attempts at only trying to confirm a single preliminary assumption in a CSA investigation as it allows the formation and evaluation of alternative hypotheses that should cover all plausible scenarios leading to the abuse allegation [53,54]. It has been argued that adopting a hypothesis-testing approach should especially decrease the risk of false-positive errors [55] given that in a CSA investigation a hypothesis of abuse is a necessary premise and may remain the only hypothesis if conscious efforts to add other hypotheses are not undertaken. Thus, in the current study, we integrated a hypothesis-testing procedure into the Avatar Training.

However, there is some tension between hypothesis-testing and the use of open-ended questions, the capstone of a good quality interview. Hypothesis-testing can be construed as a confirmatory task in which questions are formed to confirm or rule out hypotheses [38,56]. Effective hypothesis-testing, when not considering children's cognitive limitations [57,58], including the tendency of children to give affirmative responses even if they do not know the answer to the question [12,59,60], typically includes the use of close-ended questions [38,61]. These may be closed directive, option-posing questions, whilst open-ended questions are not often used since this type of question may not seem an ideal way of testing alternative hypotheses. Thus, in terms of question types, the information-acquiring process in hypothesis-testing may, to some extent, be indistinguishable from low-quality interviewing. Therefore, the challenge lies in achieving good question quality while being able to test hypotheses. An emphasis on hypothesis-testing perhaps always drives question quality down. This could especially be the case if both the hypothesis-testing and best practice in terms of question types are learned simultaneously which might cause cognitive overload for the learner [62–64]. However, it is also possible that hypothesis-testing counteracts a sole focus on a CSA hypothesis and that this will result in more open information seeking and, consequently, more use of open questions. It may also be that a combination of a hypothesis-testing approach with feedback on question types would not result in any negative effect on interview quality when measured in question types. That is, interviewers may be able to execute the hypothesis-testing process to a satisfactory level primarily through the use of open questions in this situation.

The present study was the first to bring the two approaches together. We examined not only the effect of teaching hypothesis-testing alone on the quality of CSA investigative interviewing but also the effects of a combination of hypothesis-testing and feedback on question types. The validity of a Chinese version of the Avatar Training approach was assessed as well. The present study tested the following hypotheses:

1. Hypothesis-testing (i.e. HT) was expected to improve interviewing quality (i.e. question types used), and consequently the quality of the information derived from the Avatars.

2. Feedback on outcome and question types was expected to improve interviewing quality, and consequently the quality of the information derived from the Avatars.

3. The combination of hypothesis-testing and feedback on Feedback was expected to result in more improvement in interview quality than the two interventions alone.

4. All three intervention groups were expected to perform better than the control group which received neither intervention (ie. no feedback on Feedback and no HT).

5. The total number of hypotheses formulated by the participants, the number of non-CSA hypotheses, and the ratio (more non-CSA compared to CSA hypotheses) were expected to predict both better question quality and quality of information derived from the Avatars (these analyses were limited to the HT conditions).

6. Improvement in interviewing quality was expected to mediate the impact of HT manipulation on interview quality variables (eliciting details).

We also explored how the experimental manipulations and the question types used impacted the Avatars' perceived reliability, contradictoriness and suggestibility.

## Materials and methods

### Pre-registration

The current study was pre-registered on the Open Science Framework (OSF): https://osf.io/7ytnh. After the data analysis had begun, there were a few deviations from our original analysis plan regarding hypothesis-testing variables: To tackle the ratio from non-CSA to CSA hypotheses becoming infinite resulting from the absence of CSA hypotheses, we used the ratio of the number of CSA hypotheses of the total number of hypotheses as well as the number of non-CSA hypotheses of the total number of hypotheses. Apart from this, we added a measure of the number of non-CSA hypotheses subtracted from the number of CSA hypotheses.

The main statistical models were adjusted in the following manner: First, we chose maximum likelihood estimation (MLE) with conventional standard errors in lieu of bivariate Pearson correlations to be able to assess associations between variables at both the interview and participant levels. Second, we used Cohen's $d$ effect size in lieu of Holm's method to quantify the magnitude of the differences in planned comparisons. Third, the Generalized Linear Model (GLM) included Time as a potential predictor of the correctness of conclusions in this study (All details were described in the Statistical Analyses). We also ran exploratory analyses to further examine the impact of the experimental manipulation on participants' assessment of information from avatars.

### Participants

The participants were 81 (55 women) university students ($M_{age}$ = 21.19, $SD$ = 1.70) who completed the experiment for 100 RMB. The demographic information of one participant was

missing. The conditions for recruiting the participants are: 1) 18 years old or older, 2) enrolled university student, and 3) the native language is Chinese. Informed consent was obtained from all participants via Qualtrics for inclusion in the study. Individuals that are younger than 18 years old or non-native speakers of Chinese were excluded from this research. Among these participants, no one had either experience in CSA investigations or parenting experience. Eight (9.9%) had experience taking a training course in child interviewing, and five (6.2%) had experience interviewing children. The board of research ethics at New York University Shanghai approved the study before the data collection commenced (2020–008).

## Design

The present study employed a 2 (Feedback on Outcome and Question-type (Feedback): Present vs. Not present: between subjects) * 2 (hypothesis-testing (HT): A practice of HT and HT before each interview vs. Neither practice of HT or HT before each interview: between subjects) * 5 (Time: From 1st to 5th interviews: within-subjects) mixed design. Participants were randomly allocated into the control ($n = 20$), the feedback only ($n = 19$), the hypothesis-testing only ($n = 22$), or the combination of feedback and hypothesis-testing ($n = 20$) conditions.

### Avatar training

The simulated interview application used for the experiment included avatars varying in their age (4 or 6 years old), gender (male or female), and presence of abuse (yes or no). Each condition has two avatars, resulting in a total of 16 avatars for the eight conditions. The Avatar Training adopted an answer selection algorithm designed by Pompedda et al. [26], which is based on the findings on the impact of question types on the elicited responses from children [12,48,65]. During each session, the questions asked by the interviewer were coded by the operator into the question types, and the answer selection algorithm process then chose the avatar's responses and showed the relevant video clip to the interviewer automatically.

### Answer selection

Avatar Training is equipped with an algorithm that gives responses (either correct, irrelevant, or wrong) to the questions asked by the interviewers with predefined probabilities. The predefined probabilities of giving answers to either open-ended questions or closed questions are based on empirical research on real children's behavioral patterns during interviews [26]. For example, a 4-year-old avatar's probability of giving a "yes" response to a closed question is different from that of a 6-year-old avatar, with considerations of children's cognitive development. Each avatar has nine relevant details and nine neutral details stored in their pre-existing "memory". The relevant details contain information that can either substantiate the presence of abuse or offer a non-abuse explanation for the event. The neutral details are dispensable details that are prepared to make the simulation more realistic. The avatar algorithm would provide one of these either relevant or neutral details in response to every recommended question asked based on the probability that is pre-assigned. Only when recommended questions were asked would the relevant or neutral information be provided. The presentation of relevant and neutral details followed a set order, with the last four relevant details containing the crucial contexts to find the truth of each case. Interviewers could acquire incorrect details that were in conflict with the predefined memories when they asked not recommended questions. Under this setting, the avatars provide incorrect responses to not recommended questions occasionally (e.g., giving an affirmative answer to an option-posing question when this information is absent in the avatar's memory) that mimic children 's responses to such questions in real life.

## Procedure

The experiment was carried out via WeChat and Zoom, taking approximately 2 hours for each participant. The probability and magnitude of harm/discomfort anticipated as a result of participating in this study were not greater than those ordinarily encountered in daily life or during the performance of routine physical or psychological examinations or tests. Participants were informed about the potential mental distress they might experienced in the course of the experiment and were recommended to consider carefully whether they wanted to participate in this research in the case that they had had abusive experiences since they might have more distress. After completing the informed consent and demographic information forms, participants in all four conditions read the guidelines for the correct questioning style and answered two comprehension checks (e.g., 'If the child provides a detail regarding the alleged situation, for example 'he punched me' which is the best question to ask? A *Did it hurt*? B *Who punched you*? C *Was it your father*?'). Participants were asked to read the instructions again and re-enter their answers in case they gave an incorrect answer to one or both of the comprehension checks. Participants in either the HT only condition or the HT+Feedback condition were also asked to read the guidelines for hypothesizing and practice hypothesis-testing using two mock cases.

During the interview rounds, after reading the background information of each alleged CSA case, participants in the HT condition and the HT+Feedback condition were first asked to complete the form of hypotheses regarding the presence or absence of child sexual abuse in the case (see Table 1), and then answered two questions about their preliminary impression of the case before the interview: (1) the presence of the abuse ("present" or "absent") and (2) confidence in their assessment on a 6 point scale ("50% to "100%). For participants in the Feedback only condition and the control condition, there was no hypothesis generation procedure and they were asked to answer the two questions about their preliminary belief right after reading the background information. In all conditions, participants were instructed to verbally ask questions to gather information from the avatar in order to determine the presence or absence of sexual abuse.

Each participant conducted five interviews with five avatars randomly selected from the sixteen avatars. Each interview lasted up to 10 minutes. The participants could terminate the

**Table 1. (Non) CSA hypotheses built by participants for an example mock case.**

| Example of an mock case |
|---|
| Nicholas is a 6-year-old boy who lives alone with his mother. Nicholas, who has attended elementary school for 6 months, has always had problems with submitting to the authority of teachers. He is described as a restless child who is never seated, and he argues frequently with schoolmates. The only place where Nicholas seems to be at ease is during dance lessons, which his mother pushed him to start at the age of 4. Ever since the new teacher Richard arrived at the dance school, it seems that his passion for dance has increased. Richard and Nicholas have established a very good relationship and Nicholas often goes along with other children to Richard's home for private lessons. After one of these lessons he tells his mother that he was alone with Richard and that he had danced naked with him. His mother, alarmed by this fact, contacted the teacher to ask for an explanation. Richard said that this is absolutely not true, and that it was a fantasy Nicholas had, and that Nicholas had asked him to get naked and dance, but he absolutely said no to him. |

| CSA hypothesis |
|---|
| Richard stayed together with Nicholas who asked Nicholas to get naked to dance, and abused Nicholas. |

| Non-CSA hypothesis |
|---|
| Richard asked Nicholas to get naked to dance, and Nicholas refused what Richard said. |

| Case outcomes |
|---|
| Nicholas has been actually abused by the teacher who after having stripped him, started touching him, and he explained to the child that it was a particular type of dance lesson with which he would become very capable, and that he should not tell anyone because his teammates would be jealous. |

interview before 10 min had passed if they believed they had already gained enough information from the avatar. After each interview, participants were asked to answer three questions regarding the reliability of the information from the avatar: (1) How reliable is the information given by the avatar you just interviewed? not reliable (0%)—highly reliable (100%) (2) To what extent was the information given by the avatar you just interviewed contradictory? not contradictory (0%)—highly contradictory (100%) (3) How suggestible (i.e. going along with leading questions) was the avatar you just interviewed? not suggestible (0%)—highly suggestible (100%). Participants were also asked to make judgments on the presence or the absence of child sexual abuse for the second time based on the information elicited from the avatar. Different from the preliminary assumption, participants needed to first decide the presence or absence of the abuse ("present" or "absent") and then offer a detailed account of the event. If the answer to the first question was "present", participants needed to provide an account of where, how, and by whom was the avatar abused. If the judgment was "absent", participants needed to provide an account of the unfounded CSA allegation. The participants in the Feedback condition received feedback on the case outcome and question type (two recommended questions and two not recommended questions) after each interview. The feedback on question type contained the categorizations of the questions as well as their impacts on the reliability of children's statements.

## Statistical analyses

Many of the hypotheses formulated by the participants were in the format "A was sexually abused by B" and "A was not sexually abused by B". That is, different from what we expected, participants in these cases did not actually do any in-depth analysis of the specific scenarios that could underlie the suspicion of abuse, which is essential for the hypothesis-testing approach. Hence, we reported the analyses using the original hypothesis variables (i.e., the numbers of CSA and non-CSA hypotheses) per the pre-registered plan but also examined Hypothesis 5 using the number of CSA and non-CSA hypotheses after excluding hypotheses of this unanalytical type (named: revised Hypothesis related variables).

Bland and Altman [66] introduced the correlation analysis of repeated observations. When looking at whether the increase (or decrease) of one variable within each interview is associated with the variability of the other, we removed the differences among participants and calculated within-subject correlations. When looking at whether participants with a higher (or lower) value of one variable have a trend toward having a higher (or lower) value of the other, we calculated the between-subject correlations by using the averages of the variables of each participant. Correlation analyses were modeled at both within- and between-subject levels simultaneously by using the multilevel.cor function within the R package lavaan [67]. To look at (1) the validity of the avatars' answering algorithms and (2) the associations between participants' assessment of the information from avatars and their questioning skills as well as the types of details elicited from avatars in each interview, we only focused on the within-subject level (interview level in our case) correlations coefficients to assess whether the correctness of conclusions, the types of questions, the types of details, and the assessment of the avatar in terms of reliability, contradictoriness, and suggestibility, are associated with each other. Afterward, we looked at both the within- and between-subject levels for the hypotheses related variables, types of questions, as well as the number of details to examine Hypothesis 5. The Maximum Likelihood estimation (MLE) with conventional standard errors was used as the estimator to provide the best fitted linear model. The correlation coefficients at the interview level represent the linear associations between two measures after controlling the variance between participants in our case.

Given that participants who were allocated into either the HT or the HT+Feedback conditions built hypotheses before the first interview, while the Control and the Feedback conditions did not receive any intervention before the first interview, we compared baseline performance according to whether participants built hypotheses or not using a series of *t*-tests. In addition, for exploratory purposes, we also compared differences in hypothesis-testing between the HT condition and the HT+Feedback condition.

In the main analyses, we conducted a series of 2 (Feedback: Present vs. Not present: between subjects) * 2 (Hypothesis-testing (HT): Training in HT and HT before each interview vs. Neither training in HT nor HT before each interview (between subjects) * 5 (Interview number: From 1st to 5th interviews: within-subjects) three-way mixed multivariate analysis of variance (MANOVA) to test the effect of Avatar training on the types of questions, types of details, and the assessment of information from Avatars by using the R package MANOVA. RM [68], we chose the modified ANOVA-type statistic test (MATS) with the parametric boot-strap procedure, which is suitable for non-normal multivariate models. Compared to the Wald-type statistical test (WTS), MATS provides robust test statistics without the requirement of extremely large sample sizes [69]. For the analyses at the univariate level, we used the R package afex [70] to conduct a series of 2 (Feedback: Present vs. Not present: between subjects) * 2 (Hypothesis-testing (HT): Training in HT and HT before each interview vs. Neither training in HT nor HT before each interview (between subjects) * 5 (Interview number: From 1st to 5th interviews: within-subjects) three-way mixed analysis of variance (ANOVA) to test the effect of Avatar training on the types of questions, types of details, and the assessment of information from Avatars. Mauchly's Test of Sphericity revealed that all dependent variables except the proportion of recommended questions, the number of wrong details, and the perceived contradictoriness of information from Avatars satisfied the sphericity assumption since their epsilon (ε) values were > 0.75. The Greenhouse-Geisser correction was used to correct the degrees of freedom of the proportion of recommended questions and the number of wrong details since the epsilon (ε) values was < 0.75 [71]. Planned pairwise comparisons were conducted by using the R package emmeans [72]. automatically loaded package graphics and stats [73]. Mean Differences (MD) were calculated by using the pair () function with the Tukey method to adjust for multiple comparisons to identify at which interview the differences were significant. When looking at the dichotomous variables: (1) correctness of conclusions and (2) correctness of complete conclusions, we used R package lme4 [74] to perform a generalized linear mixed model, and the 95% confidence intervals of the estimation were calculated by using the Wald method.

## Results

### Descriptive statistics

Out of 5 participants, 4 participants from the HT condition (1 participant from the HT+-Feedback condition) have prior experience interviewing children. Significant condition differences were found in both experiences in training course taking ($\chi^2(3) = 13.45$, $p = .002$, $\varphi = 0.41$) and interviewing children ($\chi^2(3) = 7.98$, $p = .046$, $\varphi = 0.32$). However, after combining these two individual variables into one variable to reflect participants' previous knowledge or experience in investigative interviews, there was no significant difference among the four conditions ($\chi^2(3) = 4.98$, $p = .187$, $\varphi = 0.25$).

Overall means in the first interview were 9.01 ($SD = 6.56$) for the number of recommended questions, 13.02 ($SD = 8.13$) for the number of non-recommended questions, 41.8% ($SD = 25.12$) for the proportion of recommended questions, 2.12 ($SD = 2.07$) for the number of relevant details, 1.89 ($SD = 1.86$) for neutral details, 3.12 ($SD = 2.53$) for wrong details, 64.80

($SD$ = 17.61) for avatar reliability, 43.09 ($SD$ = 23.25) for avatar suggestibility, 38.09 ($SD$ = 18.85) for avatar contradictoriness, 1.73 ($SD$ = 0.89) for the number of CSA hypotheses, and 1.60 ($SD$ = 0.91) for the number of non-CSA hypotheses.

Over five interviews, participants reached a correct conclusion (i.e., whether the Avatar was sexually abused or not) 58.8% of the time. However, only 14.6% of the conclusions contained a completely accurate account of what had happened indicating that the interviewing task was quite difficult. On average, the number of recommended questions used by participants in each interview ($M$ = 12.11, $SD$ = 8.83) was roughly the same as the number of non-recommended questions ($M$ = 12.38, $SD$ = 9.57): the proportion of recommended questions was 50.3% ($SD$ = 27.89). Participants on average elicited 2.71 relevant details ($SD$ = 2.39), 2.77 neutral details ($SD$ = 2.51), and 3.04 wrong details ($SD$ = 3.04). Additionally, concerning all five interviews, participants assessed avatars' reliability with the highest score ($M$ = 66.86, $SD$ = 18.25) compared to assessed avatars' contradictory ($M$ = 26,21, $SD$ = 28.88) and suggestibility ($M$ = 43.00, $SD$ = 23.56). These results did not show a clear numeric superiority of recommended questions among five interviews overall, and our participants failed to elicit a greater number of relevant details compared to neutral and wrong details from avatars.

## Correlations confirming correct functioning of the simulation

The correlations between the types of questions, the types of details elicited by participants, and the assessment of information of avatars at the within-subject level are presented in Table 2. The number of recommended questions and the proportion of recommended questions were significantly positively associated with both the number of relevant details and the number of neutral details, as well as negatively associated with the number of wrong details. These patterns were reversed for the number of non-recommended questions. These within-subject correlations confirm that the algorithms worked as expected.

## Preliminary analyses related to the experimental manipulation

A series of $t$-tests revealed no significant differences between experimental groups in their baseline performance with the exception of the number of wrong details, where we found a significant difference between the HT ($M_{HT}$ = 4.18, $SD$ = 2.17) and the HT+Feedback conditions ($M_{HT+Feedback}$ = 1.70, $SD$ = 2.03): $t(39.97)$ = 3.83, $p < .001$, 95% $CI[1.17, 3.79]$. This is probably a chance effect.

Regarding the hypotheses that were constructed by participants, we found significant differences between the HT and the HT+Feedback conditions in the revised number of non-CSA hypotheses ($M_{HT}$ = 1.09, $SD$ = 0.71 versus $M_{HT+Feedback}$ = 1.43, $SD$ = 0.68; $t(40)$ = -2.42, $p = .020$; 95% $CI[-0.63, -0.06]$) and in the difference between the revised number of CSA hypotheses and

**Table 2.** [a]**Correlations confirming correct functioning of the simulation.**

| Variables | 1 | 2 | 3 | 4 | 5 | 6 |
|---|---|---|---|---|---|---|
| The number of recommended questions | 1 | | | | | |
| The number of non-recommended questions | -0.36*** | 1 | | | | |
| The proportion of recommended questions | 0.71*** | -0.72*** | 1 | | | |
| The number of relevant details | 0.70*** | -0.37*** | 0.59*** | 1 | | |
| The number of neutral details | 0.68*** | -0.27*** | 0.51*** | 0.44*** | 1 | |
| The number of wrong details | -0.27*** | 0.70*** | -0.47*** | -0.23*** | -0.18*** | 1 |

[a]*$p < .05$

**$p < .01$

***$p < .001$.

non-CSA hypotheses ($M_{HT}$ = -0.48, $SD$ = 0.95 versus $M_{HT+Feedback}$ = -0.91, $SD$ = 0.92; $t(40)$ = 2.59, $p$ = .013; 95% $CI$[0.09, 0.77]). These results indicate that combined hypothesis-testing with feedback made participants build more non-CSA hypotheses than HT only. No other significant differences were detected in the number of CSA hypotheses (Original: $M_{HT}$ = 1.51, $SD$ = 0.71 versus $M_{HT+Feedback}$ = 1.57, $SD$ = 0.82; $t(40)$ = -0.30, $p$ = .768; 95% $CI$[-0.44, 0.33]; Revised: $M_{HT}$ = 0.61, $SD$ = 0.79 versus $M_{HT+Feedback}$ = 0.52, $SD$ = 0.64; $t(33.80)$ = 0.65, $p$ = .521; 95% $CI$[-0.19, 0.37]), in the total number of hypotheses (Original: $M_{HT}$ = 2.89, $SD$ = 1.22 versus $M_{HT+Feedback}$ = 3.02, $SD$ = 1.37; $t(40)$ = -0.41, $p$ = .682; 95% $CI$[-4.02, 2.66]; Revised: $M_{HT}$ = 1.68, $SD$ = 1.18 versus $M_{HT+Feedback}$ = 1.95, $SD$ = 0.96; $t(40)$ = -1.11, $p$ = .274; 95% $CI$[-3.57, 1.04]), or in the ratio between (non) CSA and total hypotheses (Original: $M_{HT}$ = 0.51, $SD$ = 0.06 versus $M_{HT+Feedback}$ = 0.52, $SD$ = 0.09; $t(40)$ = -0.51, $p$ = .610; 95% $CI$[-0.04, 0.02]; Revised: $M_{HT}$ = 0.42, $SD$ = 0.38 versus $M_{HT+Feedback}$ = 0.52, $SD$ = 0.38; $t(40)$ = -1.87, $p$ = .069; 95% $CI$[-0.21, 0.01]).

## Effects of Avatar training on questioning skills and interview quality

Multivariate level significant effects on the questioning skills (i.e., the number of recommended questions, the number of non-recommended questions, and the proportion of recommended questions, hereafter, abbreviated as Recommended, Not-recommended, and Proportion, respectively) were found for Feedback (MATS = 241.87, $p$ < .001), Interview number (MATS = 62.60, $p$ < .001), as well as the interaction between Feedback and Interview number (MATS = 86.62, $p$ < .001). However, we did not observe any significant effect for hypothesis-testing (i.e., HT; MATS = 19.67, $p$ = .130), the interaction between HT and Feedback (MATS = 8.89, $p$ = .382), the interaction between HT and Interview number (MATS = 3.29, $p$ = .750), as well as the three-way interaction between HT, Feedback and Interview number at the multivariate level (MATS = 11.76, $p$ = .056).

At the univariate level, a significant three-way interaction was detected between HT, Feedback, and Interview number on the number of recommended questions ($F(3.61, 277.67)$ = 3.78, $p$ = .007, $\eta_g^2$ = 0.02). There were significant main effects of Feedback (Recommended: $F(1, 77)$ = 9.85, $p$ = .002, $\eta_g^2$ = 0.08; Non-recommended: $F(1, 77)$ = 27.01, $p$ < .001, $\eta_g^2$ = 0.20; Proportion: $F(1, 77)$ = 39.68, $p$ < .001, $\eta_g^2$ = 0.23), and Interview number (Recommended: $F(3.61, 277.67)$ = 12.74, $p$ < .001, $\eta_g^2$ = 0.06; Non-recommended: $F(3.63, 279.13)$ = 4.50, $p$ = .002, $\eta_g^2$ = 0.02; Proportion: $F(3.10, 238.86)$ = 13.72, $p$ < .001, $\eta_g^2$ = 0.07) on types of questions. In addition, HT showed a significant main effect only on the number of recommended questions ($F(1, 77)$ = 5.43, $p$ = .022, $\eta_g^2$ = 0.04). Significant two-way interactions were also found between Feedback and Interview number on both the number of non-recommended questions and the proportion of recommended questions (Non-recommended: $F(3.63, 279.13)$ = 21.14, $p$ < .001, $\eta_g^2$ = 0.08; Proportion: $F(3.10, 238.86)$ = 18.99, $p$ < .001, $\eta_g^2$ = 0.10) (Fig 1).

To look at the effects of the HT and Feedback interventions, respectively, in each interview, we combined pairs of experimental conditions and conducted planned comparisons between HT conditions (i.e., the combination of HT and HT+Feedback conditions) and Non-HT conditions as well as between the Feedback conditions (i.e., the combination of Feedback and HT +Feedback conditions) and Non-feedback conditions. Compared to Non-feedback conditions, participants in Feedback conditions presented significantly more recommended questions during the 3rd, 4th, and 5th interviews, while fewer non-recommended questions in all interviews except the first one. Combined, Feedback conditions increased the proportion of presented recommended questions during the 2nd, 3rd, 4th, and 5th interviews, indicating that receiving feedback improved the questioning skills of participants over time. Compared to Non-HT conditions, receiving HT intervention made participants present fewer recommended questions during the 4th and 5th interviews (Table 3).

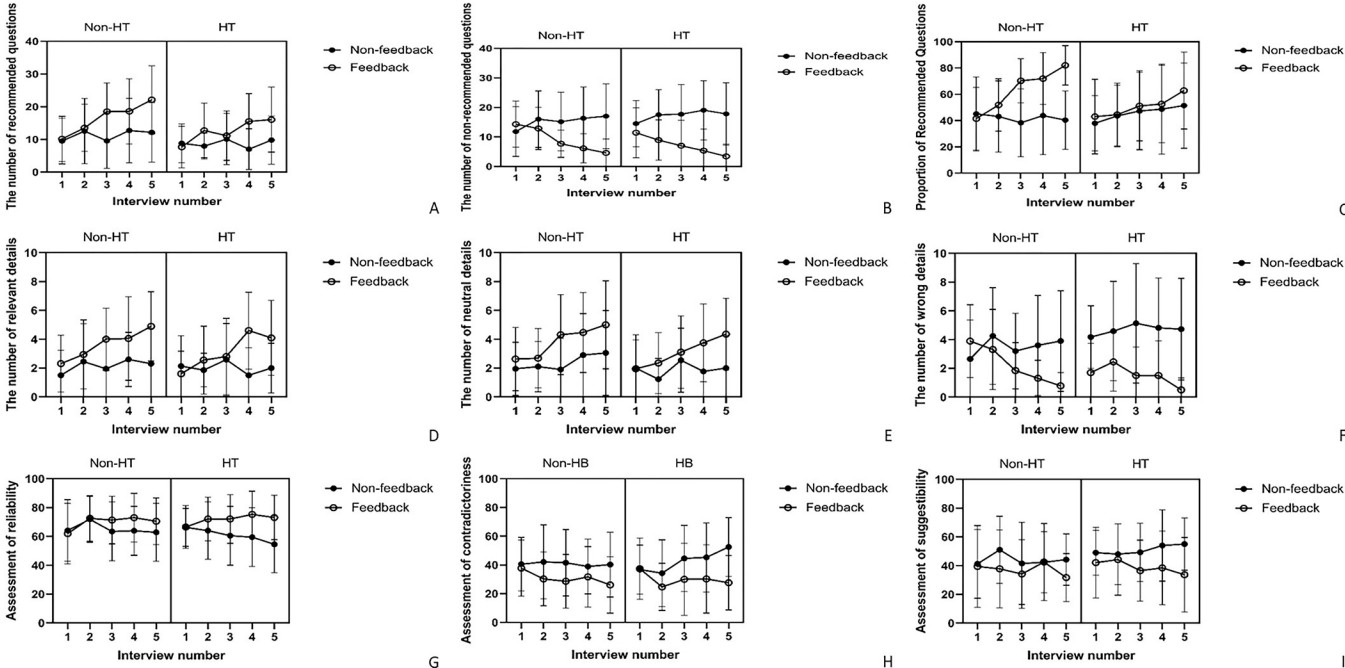

**Fig 1. Question skills, types of details, and the assessment of information from Avatars in different number of interviews among different combinations of two interventions.** HT: Participants who built hypotheses before each interview. Feedback: Participants who received feedback on outcome and question types. Panels (A): The means of recommended questions (B) the means of non-recommended questions, and (C) the mean proportions of recommended questions presented by participants in different number of interviews. Panels (D) The means of relevant details, (E) neutral details, and (F) wrong details presented by participants in different number of interviews. Panels (G) The averaged degree of reliability (H) contradictoriness, and (I) suggestibility of information from avatars assessed by participants in different number of interviews.

**Table 3. [a]The individual effect of the HT or feedback intervention on the types of questions over time.**

| Variable | Comparisons[b] | Interview number | d | SE | df | p | 95%CI |
|---|---|---|---|---|---|---|---|
| **The number of recommended questions** | Feedback vs. Non-feedback | 3 | 0.58 | 0.23 | 79 | .011 | [0.13, 1.03] |
| | Feedback vs. Non-feedback | 4 | 0.83 | 0.23 | 79 | < .001 | [0.37, 1.29] |
| | Feedback vs. Non-feedback | 5 | 0.84 | 0.23 | 79 | < .001 | [0.38, 1.30] |
| | HT vs. Non-HT | 4 | -0.46 | 0.23 | 79 | .042 | [-0.91, -0.01] |
| | HT vs. Non-HT | 5 | -0.47 | 0.23 | 79 | .037 | [-0.92, -0.02] |
| **The number of non-recommended questions** | Feedback vs. Non-feedback | 2 | -0.73 | 0.23 | 79 | .002 | [-1.19, -0.27] |
| | Feedback vs. Non-feedback | 3 | -1.06 | 0.24 | 79 | < .001 | [-1.53, -0.59] |
| | Feedback vs. Non-feedback | 4 | -1.41 | 0.25 | 79 | < .001 | [-1.90, -0.91] |
| | Feedback vs. Non-feedback | 5 | -1.63 | 0.26 | 79 | < .001 | [-2.15, -1.12] |
| **The proportion of recommended questions** | Feedback vs. Non-feedback | 2 | 0.75 | 0.23 | 79 | .001 | [0.29, 1.21] |
| | Feedback vs. Non-feedback | 3 | 1.21 | 0.24 | 79 | < .001 | [0.73, 1.70] |
| | Feedback vs. Non-feedback | 4 | 1.58 | 0.26 | 79 | < .001 | [1.08, 2.09] |
| | Feedback vs. Non-feedback | 5 | 1.90 | 0.27 | 79 | < .001 | [1.36, 2.43] |

[a]Only significant pairwise differences are presented. Non-significant pairwise differences can be found at Open Science Framework (OSF): https://osf.io/7ytnh.

[b]Feedback (i.e., the combination of Feedback and HT+Feedback conditions): Participants received feedback on Outcome and Question-types after each interview; HT (i.e., the combination of HT and HT+Feedback conditions): Participants practiced hypothesis-testing before each interview.

Planned comparisons between each pair of conditions in each number of interviews revealed that participants who were in the Feedback condition presented significantly more recommended questions and fewer non-recommended questions compared to HT during the 3rd, 4th, and 5th interviews. Compared to the Control condition, participants in the Feedback condition also presented more recommended questions during the 3rd and 5th interviews, while fewer non-recommended questions during the 3rd, 4th, and 5th interviews. Compared with HT only, combining this intervention with feedback made participants present significantly more recommended questions during the 4th interview. As expected, this significant and positive effect of receiving the combined intervention was also observed in the number of non-recommended questions compared to both HT and Control conditions during the 2nd, 3rd, 4th, and 5th interviews. Combined, regarding the proportion of recommended questions, both Feedback and HT+Feedback conditions increased the proportion of recommended questions during all interviews in comparison with both HT and Control conditions (Table 4).

For the quality of the information derived from the Avatars (i.e., the number of relevant details, the number of neutral details, and the number of wrong details, hereafter, abbreviated to Relevant, Neutral, and Wrong, respectively), multivariate level significant effects were found for Feedback (MATS = 133.99, $p < .001$), Interview Number (MATS = 60.80, $p < .001$), as well as for the interaction between Feedback and Interview number (MATS = 43.01, $p < .001$). Additionally, a significant interaction between HT, Feedback and Interview number was also observed at the multivariate level (MATS = 15.48, $p = .046$). However, we did not find any significant effect for HT (MATS = 9.29, $p = .285$), the interaction between HT and Feedback (MATS = 13.21, $p = .161$), as well as the interaction between HT and Interview number (MATS = 5.05, $p = .763$) at the multivariate level.

At the univariate level, there was a significant three-way interaction among HT, Feedback, and Interview number on the number of relevant details ($F(3.82, 294.06) = 2.93$, $p = .023$, $\eta_g^2 = 0.02$). Main effects of both the Feedback (Relevant: $F(1, 77) = 14.46$, $p < .001$, $\eta_g^2 = 0.08$; Neutral: $F(1, 77) = 13.80$, $p < .001$, $\eta_g^2 = 0.08$; Wrong: $F(1, 77) = 24.59$, $p < .001$, $\eta_g^2 = 0.15$), and Interview number (Relevant: $F(3.82, 294.06) = 8.46$, $p < .001$, $\eta_g^2 = 0.06$; Neutral: $F(3.75, 288.40) = 9.73$, $p < .001$, $\eta_g^2 = 0.06$; Wrong: $F(3.25, 250.53) = 3.50$, $p = .014$, $\eta_g^2 = 0.02$) were significant on all types of details elicited by participants. Moreover, a significant interaction between Feedback and Interview number was found on both the number of neutral ($F(3.75, 288.40) = 2.91$, $p = .025$, $\eta_g^2 = 0.02$) and wrong details ($F(3.25, 250.53) = 6.46$, $p < .001$, $\eta_g^2 = 0.04$). When looking at the number of wrong details, there was also a significant interaction between the HT and Feedback ($F(1, 77) = 4.36$, $p = .040$, $\eta_g^2 = 0.03$) (Fig 1).

Compared to Non-feedback conditions, participants in Feedback conditions significantly elicited more relevant details during the 3rd, 4th and 5th interviews. Additionally, Feedback conditions had significantly more neutral details, as well as fewer wrong details, in all the number of interviews except the first. However, no significant differences between HT and Non-HT conditions were observed (Table 5).

Planned comparisons between each pair of conditions in each number of interviews revealed that participants in the Feedback condition significantly elicited more relevant details compared to participants in the Control condition during the 3rd and 5th interviews. The Feedback also improved the number of elicited neutral details during the 3rd interview. Compared with the HT condition, combining HT with Feedback made participants elicit significantly more relevant details during the 4th and 5th interviews, while more neutral details during the 5th interview. Additionally, this combined intervention significantly increased the number of relevant details compared to the Control condition during the 4th interview. Compared to the HT condition, participants in the Feedback condition elicit more relevant and neutral details during the 4th and 5th interviews. For the number of wrong details elicited by

**Table 4. ªPlanned comparisons on the types of questions between experimental conditions over time.**

| Variable | Comparisons (Experimental condition)ᵇ | Interview number | MD | SE | df | p | 95%CI |
|---|---|---|---|---|---|---|---|
| **The number of recommended questions** | HT+Feedback vs. Feedback | 3 | -7.33 | 2.64 | 77 | .034 | [-14.25, -0.40] |
| | HT vs. Feedback | 3 | -8.44 | 2.58 | 77 | .009 | [-15.21, −1.66] |
| | Feedback vs. Control | 3 | 8.89 | 2.64 | 77 | .006 | [2.05, 15.91] |
| | HT+Feedback vs. HT | 4 | 8.80 | 2.72 | 77 | .010 | [1.65, 15.95] |
| | HT vs. Feedback | 4 | -11.58 | 2.76 | 77 | < .001 | [-18.82, -4.33] |
| | HT vs. Feedback | 5 | -12.33 | 2.81 | 77 | < .001 | [-19.71, -4.95] |
| | Feedback vs. Control | 5 | 9.96 | 2.87 | 77 | .005 | [2.41, 17.51] |
| **The number of non-recommended questions** | HT+Feedback vs. HT | 2 | -8.60 | 2.51 | 77 | .005 | [-15.20, -2.00] |
| | HT+Feedback vs. Control | 2 | -7.10 | 2.57 | 77 | .036 | [-13.85, -0.35] |
| | HT+Feedback vs. HT | 3 | -10.62 | 2.69 | 77 | < .001 | [-17.74, -3.63] |
| | HT+Feedback vs. Control | 3 | -8.15 | 2.75 | 77 | .021 | [-15.37, -0.93] |
| | HT vs. Feedback | 3 | 10.00 | 2.72 | 77 | .003 | [2.85, 17.15] |
| | Feedback vs. Control | 3 | -7.47 | 2.79 | 77 | .044 | [-14.78, -0.15] |
| | HT+Feedback vs. HT | 4 | -13.70 | 2.66 | 77 | < .001 | [-20.67, -6.72] |
| | HT+Feedback vs. Control | 4 | -10.95 | 2.72 | 77 | < .001 | [-18.09, -3.81] |
| | HT vs. Feedback | 4 | 12.94 | 2.69 | 77 | < .001 | [5.87, 20.01] |
| | Feedback vs. Control | 4 | -10.20 | 2.76 | 77 | .002 | [-17.43, -2.96] |
| | HT+Feedback vs. HT | 5 | -14.42 | 2.57 | 77 | < .001 | [-21.17, -7.66] |
| | HT+Feedback vs. Control | 5 | -13.60 | 2.63 | 77 | < .001 | [-20.51, -6.69] |
| | HT vs. Feedback | 5 | 13.24 | 2.61 | 77 | < .001 | [6.39, 20.08] |
| | Feedback vs. Control | 5 | -12.42 | 2.67 | 77 | < .001 | [-19.42, -5.42] |
| **The proportion of recommended questions** | HT+Feedback vs. HT | 2 | 23.95 | 6.82 | 77 | .004 | [6.05, 41.86] |
| | HT vs. Feedback | 2 | -19.35 | 6.91 | 77 | .032 | [-37.50, -1.20] |
| | HT+Feedback vs. HT | 3 | 25.31 | 7.30 | 77 | .005 | [6.13, 44.48] |
| | HT+Feedback vs. Control | 3 | 24.05 | 7.47 | 77 | .010 | [4.42, 43.68] |
| | HT vs.Feedback | 3 | -33.17 | 7.40 | 77 | < .001 | [-52.61, -13.73] |
| | Feedback vs. Control | 3 | 31.91 | 7.57 | 77 | < .001 | [12.03, 51.80] |
| | HT+Feedback vs. HT | 4 | 46.73 | 7.26 | 77 | < .001 | [27.67, 65.79] |
| | HT+Feedback vs. Control | 4 | 31.25 | 7.43 | 77 | < .001 | [11.74, 50.76] |
| | HT vs. Feedback | 4 | -43.63 | 7.36 | 77 | < .001 | [-62.95, -24.31] |
| | Feedback vs. Control | 4 | 28.15 | 7.53 | 77 | .002 | [8.38, 47.91] |
| | HT+Feedback vs. HT | 5 | 40.08 | 6.72 | 77 | < .001 | [22.43, 57.73] |
| | HT+Feedback vs. Control | 5 | 37.55 | 6.88 | 77 | < .001 | [19.49, 55.61] |
| | HT vs. Feedback | 5 | -44.23 | 6.81 | 77 | < .001 | [-62.12, -26.35] |
| | Feedback vs. Control | 5 | 41.70 | 6.97 | 77 | < .001 | [23.40, 60.00] |

ªOnly significant pairwise differences are presented. Non-significant pairwise differences can be found at Open Science Framework (OSF): https://osf.io/7ytnh.
ᵇControl: Participants neither practiced hypothesis-testing before each interview nor receive feedback on Outcome and Question-types; HT: Participants practiced hypothesis-testing before each interview; Feedback: Participants received feedback on Outcome and Question-types after each interview; HT+Feedback: Participants practiced hypothesis-testing before each interview as well as receiving feedback on Outcome and Question-types after each interview.

participants, compared to HT condition, receiving Feedback intervention resulted in significantly fewer wrong details presented by the participants during the 3rd, 4th, and 5th interviews. Participants in the Feedback condition also elicited fewer wrong details during the 5th interview in comparison with participants in the Control condition. Moreover, compared with HT only, combining the HT with Feedback intervention significantly reduced the number of wrong details during the 3rd, 4th, and 5th interviews. Also, the combined intervention made

**Table 5. [a]The individual effect of the HT or feedback intervention on the types of details over time.**

| Variable | Comparisons[b] | Interview number | d | SE | df | p | 95%CI |
|---|---|---|---|---|---|---|---|
| **The number of relevant details** | Feedback vs. Non-feedback | 3 | 0.46 | 0.23 | 79 | .043 | [0.01, 0.91] |
| | Feedback vs. Non-feedback | 4 | 0.97 | 0.24 | 79 | < .001 | [0.50, 1.44] |
| | Feedback vs. Non-feedback | 5 | 1.03 | 0.24 | 79 | < .001 | [0.56, 1.50] |
| **The number of neutral details** | Feedback vs. Non-feedback | 2 | 0.47 | 0.23 | 79 | .038 | [0.02, 0.92] |
| | Feedback vs. Non-feedback | 3 | 0.60 | 0.23 | 79 | .009 | [0.14, 1.05] |
| | Feedback vs. Non-feedback | 4 | 0.69 | 0.23 | 79 | .003 | [0.24, 1.15] |
| | Feedback vs. Non-feedback | 5 | 0.80 | 0.23 | 79 | < .001 | [0.34, 1.26] |
| **The number of wrong details** | Feedback vs. Non-feedback | 2 | -0.53 | 0.23 | 79 | .021 | [-0.98, -0.07] |
| | Feedback vs. Non-feedback | 3 | -0.87 | 0.23 | 79 | < .001 | [-1.33, -0.41] |
| | Feedback vs. Non-feedback | 4 | -1.00 | 0.24 | 79 | < .001 | [-1.47, -0.53] |
| | Feedback vs. Non-feedback | 5 | -1.42 | 0.25 | 79 | < .001 | [-1.92, -0.93] |

[a]Only significant pairwise differences are presented. Non-significant pairwise differences can be found at Open Science Framework (OSF): https://osf.io/7ytnh.

[b]Feedback (i.e., the combination of Feedback and HT+Feedback conditions): Participants received feedback on Outcome and Question-types after each interview.

participants elicit significantly fewer wrong details during the 5th interviews in comparison with the Control condition. In sum, these results illustrated the importance of Feedback to improve the interview quality by eliciting more relevant and neutral details, as well as fewer wrong details as a function of the number of interviews (Table 6).

In sum, Feedback intervention (rather than HT intervention) improved the interviewing quality, and the quality of the information derived from avatars, these results did not support Hypothesis 1 but provided support to Hypothesis 2. Compared to the HT condition (rather than the Feedback condition), the combination of HT and Feedback resulted in more improvement in interview quality. Hence, Hypothesis 3 was partially supported. Since the hypothesis-testing intervention did not have the expected effect, Hypothesis 5 was not supported. Due to the effect of HT not being found, we were not able to perform a mediation analysis to test Hypothesis 6 concerning the potential mediation role of improvement in questioning skills on the impact of hypothesis-testing intervention on interview quality variables.

## Effects of Avatar training on the assessment of the avatars

For the assessment of the Avatars (i.e., perceived reliability, contradictoriness, and suggestibility of the information from the avatars, hereafter, abbreviated as Reliability, Contradictoriness, and Suggestibility, respectively), multivariate level significant effects were found of the Feedback (MATS = 60.56, $p < .001$), and the interaction between Feedback and Interview number (MATS = 20.69, $p = .011$). However, no significant effect was observed for the HT (MATS = 3.87, $p = .594$), Interview number (MATS = 9.76, $p = .272$), the interaction between HT and Feedback (MATS = 4.49, $p = .544$), the interaction between HT and Interview number (MATS = 7.03, $p = .529$, as well as the three-way interaction between HT, Feedback, and Interview number at the multivariate level (MATS = 5.11, $p = .749$).

At the univariate level, there was no significant three-way interaction between HT, Feedback, and Interview number (Reliability: $F(3.68, 264.77) = 0.21$, $p = .918$, $\eta_g^2 < 0.01$; Contradictoriness: $F(3.58, 258.05) = 0.60$, $p = .642$, $\eta_g^2 < 0.01$; Suggestibility: $F(3.68, 265.32) = 1.34$, $p = .259$, $\eta_g^2 = 0.01$). There was no significant main effect of Interview number (Reliability: $F(3.68, 264.77) = 1.44$, $p = .223$, $\eta_g^2 = 0.01$; Contradictoriness: $F(3.58, 258.05) = 0.96$, $p = .425$, $\eta_g^2 = 0.01$; Suggestibility: $F(3.68, 265.32) = 1.07$, $p = .370$, $\eta_g^2 = 0.01$), but a significant main effect of Feedback was found (Reliability: $F(1, 72) = 9.36$, $p = .003$, $\eta_g^2 = 0.05$;

**Table 6. [a]Planned comparisons on the types of details between experimental conditions over time.**

| Variable | Comparisons (Experimental condition)[b] | Interview number | MD | SE | df | p | 95%CI |
|---|---|---|---|---|---|---|---|
| **The number of relevant details** | Feedback vs. Control | 3 | 2.05 | 0.76 | 77 | .043 | [0.05, 4.05] |
| | HT+Feedback vs. HT | 4 | 3.10 | 0.73 | 77 | < .001 | [1.18, 5.02] |
| | HT+Feedback vs. Control | 4 | 2.00 | 0.75 | 77 | .044 | [0.04, 3.96] |
| | HT vs. Feedback | 4 | -2.55 | 0.74 | 77 | .005 | [-4.50, -0.61] |
| | HT+Feedback vs. HT | 5 | 2.10 | 0.71 | 77 | .020 | [0.24, 3.96] |
| | HT vs. Feedback | 5 | –2.90 | 0.72 | 77 | < .001 | [-4.78, -1.01] |
| | Feedback vs. Control | 5 | 2.60 | 0.73 | 77 | .004 | [0.67, 4.52] |
| **The number of neutral details** | Feedback vs. Control | 3 | 2.42 | 0.78 | 77 | .014 | [0.38, 4.45] |
| | HT vs. Feedback | 4 | -2.70 | 0.81 | 77 | .007 | [-4.83, -0.58] |
| | HT+Feedback vs. HT | 5 | 2.35 | 0.83 | 77 | .030 | [0.16, 4.54] |
| | HT vs. Feedback | 5 | 3.00 | 0.84 | 77 | .004 | [-5.22, -0.78] |
| **The number of wrong details** | HT+Feedback vs. HT | 3 | -3.64 | 0.89 | 77 | < .001 | [-5.97, -1.30] |
| | HT vs. Feedback | 3 | 3.29 | 0.90 | 77 | .003 | [0.93, 5.66] |
| | HT+Feedback vs. HT | 4 | -3.32 | 0.88 | 77 | .002 | [-5.62, -1.02] |
| | HT vs. Feedback | 4 | 3.50 | 0.89 | 77 | .001 | [1.17, 5.84] |
| | HT+Feedback vs. HT | 5 | -4.23 | 0.81 | 77 | < .001 | [-6.34, -2.11] |
| | HT+Feedback vs. Control | 5 | -3.40 | 0.83 | 77 | < .001 | [-5.57, -1.23] |
| | HT vs. Feedback | 5 | 3.94 | 0.82 | 77 | < .001 | [1.79, 6.08] |
| | Feedback vs. Control | 5 | -3.11 | 0.84 | 77 | .002 | [-5.31, -0.92] |

[a]Only significant pairwise differences are presented. Non-significant pairwise differences can be found at Open Science Framework (OSF): https://osf.io/7ytnh.
[b]Control: Participants neither practiced hypothesis-testing before each interview nor receive feedback on Outcome and Question-types; HT: Participants practiced hypothesis-testing before each interview; Feedback: Participants received feedback on Outcome and Question-types after each interview; HT+Feedback: Participants practiced hypothesis-testing before each interview as well as receiving feedback on Outcome and Question-types after each interview.

Contradictoriness: $F(1, 72) = 11.40$, $p = .001$, $\eta_g^2 = 0.07$; Suggestibility: $F(1, 72) = 5.75$, $p = .019$, $\eta_g^2 = 0.04$). Moreover, a significant interaction between Feedback and Interview number was found on both perceived reliability ($F(3.68, 264.77) = 2.77$, $p = .032$, $\eta_g^2 = 0.02$) and contradictoriness ($F(3.58, 258.05) = 2.83$, $p = .030$, $\eta_g^2 = 0.02$), but not the perceived suggestibility of information from avatars ($F(3.68, 265.32) = 1.35$, $p = .253$, $\eta_g^2 = 0.01$) (Fig 1).

Compared to Non-feedback conditions, participants in Feedback conditions perceived the information as more reliable during the 3rd, 4th, and 5th interviews, while as less contradictory during all interviews except the first one. Also, receiving Feedback intervention made participants perceive the avatars as less suggestible than participants in Non-feedback conditions during the 2nd, 3rd, 4th, and 5th interviews. However, there was no significant difference between HT and Non-HT conditions in the assessment of information (Table 7).

Planned comparisons between each pair of conditions in each number of interviews revealed that both Feedback and HT+Feedback conditions made participants more reliability, while less contradictoriness and suggestibility of perceived information from avatars in comparison with the HT condition during 5th interview. We also found that participants who were in the HT+Feedback condition reported more perceived reliability of the information compared to the HT condition during the 4th interview (Table 8).

## Effects of Avatar training on the correctness of conclusions

There were two variables related to correct conclusions reached by participants including (1) whether the abuse was present or not (i.e., binary choices) and (2) correctness of complete

**Table 7.** [a]The individual effect of the HT or feedback intervention on the assessment of information from avatars over time.

| Variable | Comparisons[b] | Interview number | d | SE | df | p | 95%CI |
|---|---|---|---|---|---|---|---|
| **Reliability** | Feedback vs. Non-feedback | 3 | 0.53 | 0.23 | 74 | .023 | [0.07, 1.00] |
| | Feedback vs. Non-feedback | 4 | 0.71 | 0.24 | 74 | .003 | [0.24, 1.18] |
| | Feedback vs. Non-feedback | 5 | 0.74 | 0.24 | 74 | .002 | [0.27, 1.21] |
| **Contradictoriness** | Feedback vs. Non-feedback | 2 | -0.50 | 0.23 | 74 | .033 | [-0.96, -0.03] |
| | Feedback vs. Non-feedback | 3 | -0.61 | 0.24 | 74 | .009 | [-1.08, -0.15] |
| | Feedback vs. Non-feedback | 4 | -0.51 | 0.23 | 74 | .029 | [-0.98, -0.05] |
| | Feedback vs. Non-feedback | 5 | -0.95 | 0.24 | 74 | < .001 | [-1.43, -0.47] |
| **Suggestibility** | Feedback vs. Non-feedback | 5 | -0.85 | 0.24 | 74 | < .001 | [-1.32, -0.37] |

[a]Only significant pairwise differences are presented. Non-significant pairwise differences can be found at Open Science Framework (OSF): https://osf.io/7ytnh.
[b]Feedback (i.e., the combination of Feedback and HT+Feedback conditions): Participants received feedback on Outcome and Question-types after each interview.

accounts of what happened to the Avatar (i.e., completely correct conclusions). The generalized linear mixed model for binary choice revealed that only the interaction between HT and Interview number was a significant predictor ($OR$ = 1.69, SE = 0.23, 95% $CI$ [1.09, 2.64]). More specifically, for conditions where participants did not build hypotheses before each interview, Interview number was a positive predictor of the correctness of binary choices ($OR$ = 1.35, $SE$ = 0.37, 95% $CI$ [1.08, 1.69]). For conditions with hypothesis-testing, Interview number was not a significant predictor of conclusion correctness ($OR$ = 0.96, $SE$ = 0.11, 95% $CI$ [0.79, 1.17]). For the modeling predicting the correctness of the complete account, none of the predictors in the model was statistically significant. Hence, these results did not support Hypothesis 4.

## Correlations between hypotheses testing, types of questions, and types of details

Turning to the correlations between built hypotheses and types of questions, as well as types of elicited details at both within-subject and between-subject levels are presented in Table 9. At the within-subject level, the total number of hypotheses (without excluding non-analytic hypotheses) was negatively associated with the number of recommended questions, the

**Table 8.** [a, c]Planned comparisons on the assessment of information from avatars between experimental conditions over time.

| Variable | Comparisons (Experimental condition)[b] | Interview number | MD | SE | df | p | 95%CI |
|---|---|---|---|---|---|---|---|
| **Reliability** | HT+Feedback vs. HT | 4 | 15.76 | 5.67 | 72 | .034 | [0.85, 30.68] |
| | HT+Feedback vs. HT | 5 | 18.66 | 5.76 | 72 | .010 | [3.51, 33.80] |
| | HT vs. Feedback | 5 | -16.06 | 5.84 | 72 | .037 | [-31.42, -0.70] |
| **Contradictoriness** | HT+Feedback vs. HT | 5 | -24.87 | 6.55 | 72 | .002 | [-42.09, -7.64] |
| | HT vs. Feedback | 5 | 26.39 | 6.64 | 72 | < .001 | [8.92, 43.86] |
| **Suggestibility** | HT+Feedback vs. HT | 5 | -21.32 | 6.41 | 72 | .007 | [-38.17, -4.46] |
| | HT vs. Feedback | 5 | 23.33 | 6.50 | 72 | .003 | [6.24, 40.43] |

[a]Only significant pairwise differences are presented. Non-significant pairwise differences can be found at Open Science Framework (OSF): https://osf.io/7ytnh.
[b]Control: Participants neither practiced hypothesis-testing before each interview nor received feedback on Outcome and Question-types; HT: Participants practiced hypothesis-testing before each interview; Feedback: Participants received feedback on Outcome and Question-types after each interview; HT+Feedback: Participants practiced hypothesis-testing before each interview as well as receiving feedback on Outcome and Question-types after it.
[c]Five individual cases missed the assessment of information from Avatars. Analyses were performed based on the sample size $n$ = 76.

**Table 9.** [a, f]**Within-subject and between-subject correlations between types of questions, types of details, and hypotheses built.**

| Variables | 1 | 2 | 3 | 4 | 5 | 6 | 7 | 8 | 9 | 10 | 11 |
|---|---|---|---|---|---|---|---|---|---|---|---|
| **The number of recommended questions** | 1 | -0.23 | 0.70*** | 0.94*** | 0.96*** | -0.25* | 0.12 [0.23] | 0.20 [0.43*] | 0.17 [0.44*] | -0.32 [0.08] | -0.12 [-0.22] |
| **The number of non-recommended questions** | -0.36*** | 1 | -0.81*** | -0.31* | -0.27* | 0.96*** | -0.08 [0.06] | -0.02 [-0.22] | -0.04 [-0.10] | -0.67 [0.03] | -0.20 [0.26] |
| **The proportion of recommended questions** | 0.71*** | -0.72*** | 1 | 0.77*** | 0.73*** | -0.81*** | 0.11 [0.08] | 0.14 [0.42*] | 0.13 [0.33] | 0.76 [-0.04] | -0.01 [-0.33] |
| **The number of relevant details** | 0.70*** | -0.37*** | 0.59*** | 1 | >0.99*** | -0.33* | 0.11 [0.17] | 0.21 [0.43*] | 0.17 [0.41] | -0.15 [-0.77] | -0.17 [-0.27] |
| **The number of neutral details** | 0.68*** | -0.27*** | 0.51*** | 0.44*** | 1 | -0.28* | -0.08 [0.01] | 0.07 [0.24] | <0.01 [0.19] | -0.63 [<0.01] | -0.41 [-0.22] |
| **The number of wrong details** | -0.27*** | 0.70*** | -0.47*** | -0.23*** | -0.18*** | 1 | -0.05 [0.05] | -0.03 [-0.30] | -0.04 [-0.17] | -0.49 [0.19] | -0.10 [0.32] |
| **The number of non-analytic CSA hypotheses [The number of analytic CSA hypotheses]**[b, e] | -0.11 [0.04] | 0.15 [0.07] | -0.16* [-0.05] | -0.11 [-0.02] | -0.15* [-0.09] | 0.08 [0.10] | 1 | 0.97*** [0.37] | 0.99*** [0.77***] | 0.86 [1.30] | 0.79*** [-0.48**] |
| **The number of non-analytic non-CSA hypotheses [The number of analytic non-CSA hypotheses]**[b, e] | -0.20** [-0.16*] | 0.06 [-0.03] | -0.16* [-0.08] | -0.06 [-0.09] | -0.20** [-0.08] | 0.09 [0.05] | 0.33*** [0.03] | 1 | 0.99*** [0.88***] | 0.66 [0.48] | 0.61 [-0.64***] |
| **The total number of non-analytic hypotheses [The total number of analytic hypotheses]**[b, e] | -0.19* [-0.07] | 0.11 [0.02] | -0.18* [-0.08] | -0.10 [-0.07] | -0.21** [-0.11] | 0.08 [0.09] | 0.79*** [0.72***] | 0.81*** [0.70***] | 1 | 0.69 [1.06] | 0.70* [-0.20] |
| **A ratio of non-analytic (non-) CSA hypotheses to total hypotheses [A ratio of analytic (non-) CSA hypotheses to total hypotheses]** [b, c, e] | -0.06 [-0.02] | -0.01 [0.03] | -0.08 [-0.04] | -0.10 [0.11] | -0.04 [0.07] | 0.09 [-0.07] | 0.09 [-0.02] | 0.06 [0.08] | 0.04 [0.05] | 1 | 1.09 [0.62] |
| **11. The differential between the number of non-analytic CSA and non-CSA hypotheses [The differential between the number of analytic CSA and non-CSA hypotheses]**[b, d, e] | 0.08 [0.14] | 0.08 [0.07] | <0.01 [0.02] | -0.05 [0.05] | 0.04 [-0.01] | <0.01 [0.04] | 0.59*** [0.72***] | -0.57*** [-0.67***] | <0.01 [0.04] | 0.02 [-0.07] | 1 |

[a]The lower triangular of the matrix presents the within-subject correlation coefficients and the upper triangular of the matrix presents the between-group correlation coefficients.

[b]Based on which condition participants were randomly allocated in, only 42 (81 in total) participants constructed hypotheses before conducting each interview. Hence, the correlations involved in Hypotheses (i.e., items 7, 8, 9, and 10) were analyzed based on the sample size $n(within-subject)$ = 210 and $n(between-subject)$ = 42.

[c]A ratio of (non-) CSA hypotheses to total hypotheses: Calculated by the number of (non-) CSA hypotheses divided by the total number of hypotheses, based on the presence (or absence) of the sexual abuse (e.g., if the sexual abuse was presented, A ratio of CSA hypotheses = CSA hypotheses/Total hypotheses). If participants did not construct any hypothesis before the interview, we coded the ratio as 0.

[d]The differential between the number of CSA and non-CSA hypotheses: Calculated by the number of CSA hypotheses subtracting the number of non-CSA hypotheses. A positive number represents participants who built more CSA hypotheses.

[e]Three individual cases missed the number of (non-) CSA hypotheses in the fifth interview. Mean imputation was used to fill in the missing value.

[f]*$p < .05$

**$p < .01$

***$p < .001$.

proportion of recommended questions, and the number of neutral details. More specifically, the number of both non-analytic CSA and non-CSA hypotheses showed similar patterns. These results suggest that the more hypotheses the interviewers formed, the worse the quality of the interview was.

After excluding the non-analytic hypotheses, the negative association still remained only between the number of non-CSA hypotheses and the number of recommended questions. However, a different pattern was found at the between-subject level: participants who were able to form more analytic non-CSA hypotheses asked more recommended questions and elicited more relevant details compared to participants who formed fewer non-CSA hypotheses.

**Table 10.** [b]Within-subject correlations between correctness of conclusions, types of questions, types of details, and assessment of avatars.

| Variables | 1 | 2 | 3 | 4 | 5 | 6 | 7 | 8 | 9 | 10 | 11 |
|---|---|---|---|---|---|---|---|---|---|---|---|
| Correctness of conclusions | 1 | | | | | | | | | | |
| Correctness of complete conclusions | 0.33*** | 1 | | | | | | | | | |
| The number of recommended questions | 0.07 | 0.17** | 1 | | | | | | | | |
| The number of non-recommended questions | -0.11 | -0.13* | -0.36*** | 1 | | | | | | | |
| The proportion of recommended questions | -0.07 | 0.14** | 0.71*** | -0.72*** | 1 | | | | | | |
| The number of relevant details | 0.12* | 0.30*** | 0.70*** | -0.37*** | 0.59*** | 1 | | | | | |
| The number of neutral details | 0.04 | 0.02 | 0.68*** | -0.27*** | 0.51*** | 0.44*** | 1 | | | | |
| The number of wrong details | -0.05 | 0.01 | -0.27*** | 0.70*** | -0.47*** | -0.23*** | -0.18*** | 1 | | | |
| Assessment of reliability[a] | 0.04 | 0.04 | 0.18** | -0.23*** | 0.27*** | 0.20*** | 0.17** | -0.09 | 1 | | |
| Assessment of suggestibility[a] | 0.10 | 0.05 | -0.06 | 0.10 | -0.13* | -0.12* | -0.09 | 0.16** | -0.11* | 1 | |
| Assessment of contradictoriness[a] | -0.06 | -0.06 | -0.10 | 0.31*** | -0.23*** | -0.13* | -0.08 | 0.22*** | -0.57*** | 0.21*** | 1 |

[a]Five individual cases missed the assessment of the avatars. Correlations related to the assessment of information from avatars were computed based on the sample size *n (within-subject)* = 380.

[b]*p* < .05

**p* < .01

****p* < .001.

This suggests that participants with a greater ability to form analytic hypotheses performed better at interviewing compared with their peers who were less capable of forming hypotheses. However, at the interview level, the more hypotheses, including the non-analytic ones, the participant formed, the worse the interview performance was.

## Correlations between interview quality and perception of avatars

As for the associations between interview quality and perceptions of avatars in each interview (Table 10), within-subject correlations showed that the number (and proportion) of recommended questions were positively associated with the perceived reliability of information from avatars. In addition, the proportion of recommended questions was also negatively associated with perceived contradictoriness as well as perceived suggestibility. The number of non-recommended questions was positively associated with contradictoriness while negatively associated with perceived reliability. When looking at the details elicited by participants, the number of relevant details was positively associated with perceived reliability, negatively associated with perceived suggestibility as well as perceived contradictoriness. Neutral details only had a significant positive correlation with perceived reliability. The number of wrong details had significant positive associations with perceived suggestibility and contradictoriness. In sum, these results suggest that interviewers were able to form a generally accurate appraisal of interview quality consistent with the more objective indicators such as the number of recommended questions or the number of relevant details.

## Correlations between interview quality and correctness of conclusions

Notably, when looking at the correctness in concluding the presence (or absence) of the abuse and, the correctness of complete conclusions (Table 10), there was only a statistically significant positive correlation with the number of relevant details at the within-subject level, indicating that in each interview, eliciting more relevant details resulted in the given participant being more able to reach the correct conclusion. Besides the binary choices, participants were asked to offer complete accounts of what happened to the avatars. As expected, the number (and

proportion) of recommended questions, as well as the number of relevant details were positively associated with the correctness of complete conclusions, whereas the number of non-recommended questions showed a reversed pattern. These results illustrated that the more recommended questions were presented or relevant details were elicited within each interview, a given participant was more likely to reach an accurate account of the abuse allegation.

## Discussion

The current study was the first to examine the efficacy of Avatar Training on improving interview quality in the Chinese context. Consistent with previous findings [26–28,30,33], feedback (combined on question types and case outcomes) showed positive effects on both the questioning skills of the interviewers and the amount of elicited information from avatars, this latter finding being a function of the algorithms driving the avatars. Apart from this, our main focus was to look at the effects of hypothesis-testing on interviewing behavior. Contrary to our prediction, building hypotheses before each interview did not show the expected positive effect on interview quality. However, we did notice that the incorporation of hypothesis-testing into Avatar Training did not bring about negative effects on the effectiveness of feedback. Finally, the current study made further contribution to CSA interviewing research by examining interviewer's perceptions concerning avatars' reliability and their associations with questioning skills as well as information elicited. In sum, lack of support was found for the below hypotheses in the current study:

Hypothesis 1. HT was expected to improve interviewing quality (i.e. question types used), and consequently the quality of the information derived from the Avatars.

Hypothesis 4. All three intervention groups were expected to perform better than the control group which received neither intervention (ie. no feedback on Feedback and no HT).

Hypothesis 5. The total number of hypotheses formulated by the participants, the number of non-CSA hypotheses, and the ratio (more non-CSA compared to CSA hypotheses) were expected to predict both better question quality and quality of information derived from the Avatars (these analyses were limited to the HT conditions).

Hypothesis 6. Improvement in interviewing quality was expected to mediate the impact of HT manipulation on interview quality variables (eliciting details).

On the other hand, Hypothesis 2 (Feedback on Feedback (i.e. outcome and question types) was expected to improve interviewing quality, and consequently the quality of the information derived from the Avatars.) was supported. Hypothesis 3 (The combination of hypothesis-testing and feedback on Feedback was expected to result in more improvement in interview quality than the two interventions alone.) was partially supported. In the following sections, we briefly discuss the findings, their implications, and the limitations of the current research.

### Avatar training with feedback as a reliable tool to improve interview quality

As noted above, we found that providing feedback significantly improved questioning skills, which led to better information gathering, that is, more relevant and neutral details and fewer wrong details being elicited from the Avatars. Considering that interviewing children who were allegedly sexually abused is a complex skill, training does not necessarily result in sustained improvements in the absence of continuous and timely feedback [17,25,75,76]. Our results again confirm that feedback is essential to shaping interviewer behavior. These results also mean that Avatar Training, accompanied by feedback on interviewers' behaviors and case outcomes, can be administered via the Internet and that the effect is essentially the same also in this Chinese version of Avatar Training.

Regarding the correctness of the conclusions drawn by the participants, we found that participants who presented more recommended questions made more completely correct conclusions. Using recommended questions elicits more useful details from the avatars making it possible to formulate complete accounts of what happened to the avatar. However, the interventions did not significantly influence the correctness of the conclusions. This result was inconsistent with previous Avatar Training studies. Previous studies have demonstrated that feedback positively influenced the probability of correct conclusions [26–30,33]. For example, Pompedda et al. [30] found that the percentage of groups receiving both feedback (outcome and process) was higher (29%) than the group receiving feedback on process (21%), the group receiving feedback on the outcome (15%), and the control group (17%). Compared to the 20% of completely correct conclusions obtained in the study of Pompedda et al. [30], only 14.8% arrived at completely correct conclusions in the current study. The percentage of the group receiving feedback alone was higher (18.9%) than the group receiving the hypothesis-building intervention (10%), and the group receiving both interventions (12%). The percentage of the control group was 18%. These differences were not significant.

The investigative interview is a complex practical skill that places high demands on interviewers [77]. When faced with a challenging task, decision-makers tend to cost as little cognitive effort as possible and opt for a simplifying way to make decisions [78,79]. That is to say, participants might make judgments too fast without analyzing the gathered information comprehensively, hence undermining the correctness of conclusions. In the current study, interviewers received feedback on both outcomes of the case and the question types they presented, but no information on the accuracy of details. This feedback intervention improved interview quality (i.e., interviewers presented more recommended questions). However, it may not have established a connection between the elicited details and the correctness of conclusions. A potentially useful way to meet the optimal outcome is to provide feedback regarding whether the details elicited from avatars were accurate or not to assist CSA interviewers in drawing correct conclusions in future studies.

## Hypothesis-testing: A not-yet-successful implementation

The idea behind the hypothesis-testing approach, as elaborated in Korkman et al. [50], is to combat confirmation bias by building alternative hypotheses using information related to the case at hand and then evaluating these hypotheses against information obtained from the interview and other sources. Contrary to our prediction, building hypotheses alone did not have a positive effect on questioning skills, types of elicited information, or the ability to conclude whether avatars had been sexually abused. Summed over interviews, compared to participants who did not receive any intervention, we found a non-significant trend that building hypotheses prior to the interview promoted the use of non-recommended questions (Control versus HT: $M = 15.25$, $SD = 8.68$ versus $M = 17.31$, $SD = 7.95$) and interviewers elicited more wrong details from Avatars in the HT only condition (HT: $M = 4.69$, $SD = 2.55$) than the control condition (Control: $M = 3.52$, $SD = 2.35$). While the simulated training with feedback is meant to promote a child-led interview, the intervention of hypothesis-testing requires interviewers to take on a more active role. As mentioned before, there is potentially some tension between hypothesis-testing and the use of open questions. It is natural for interviewers to ask closed questions because that may help with the effective hypothesis-testing when the interviewee's age and cognitive capacity have been taken into consideration. On the other hand, non recommended question dominated interview, which represents interviews conducted by the hypothesis-testing group (see Fig 1 Panel C), is distant from ideal for conducting a high-quality interview with young children according to existing studies [48,49,80].

However, the current results do not necessarily mean that hypothesis-testing per se lacks efficacy. In our experiment, we did not assess whether the actual questions asked tested both abuse and non-abuse hypotheses in the hypothesis-testing conditions. It is thus possible that the HT group did conduct an interview that tested hypotheses in a balanced manner albeit not using open questions. So far no studies have assessed the efficacy of such an approach to CSA interviewing. However, we did not find any differences between the groups in conclusion correctness suggesting that this approach was neither clearly superior nor inferior to the other experimental groups. Moreover, in the present study, the interviewers lacked relevant experience in CSA interviews and the hypothesis-testing approach. Previous research evaluating professionals' and laypersons' knowledge of interviewing children pointed out that students are less knowledgeable than professionals, especially uninformed when considering the negative impact of leading questions [81]. However, in the context of CSA investigative interviewing, there is no convincing evidence of the positive effects of experience alone on interview quality [28,35,46]. Therefore, we refrain from making strong claims that the current results are due to inexperience.

Notwithstanding a brief guideline on the importance of only employing recommended (e.g., open-ended questions) and non-leading questioning techniques provided to the participants before the first interview, it seemed that only building hypotheses may unintentionally shape the questioning styles of interviewers in a negative direction leading to an increased number of non-recommended questions [44,82,83]. We were also interested in understanding whether learning questioning skills and the hypothesis-testing approach simultaneously introduced too much new information to the interviewers having a negative impact on their learning outcomes. However, when combined with feedback, the negative effect of hypothesis-testing on questioning skills and interviewing quality was mitigated. These findings support the notion that feedback not only re-enforces correct questioning style, but also provides interviewers with the opportunity to continuously assess and improve their questioning style.

The combined interventions group obtained similar numbers of relevant, neutral, and wrong details as the Feedback group, but reached fewer completely correct conclusions. The tasks between the feedback group and the combined interventions group were different. One possible explanation is that when giving an explanation of what happened in the cases, the feedback group provided a report based on the information the avatar shared with them, whilst the combined intervention group had to compare the information elicited from the avatar with their hypotheses before they arrive at a judgment. It may be that information elicited was insufficient to allow for differentiating between the hypotheses and reaching a coherent conclusion.

Interestingly, in correlation analyses, we found that participants who built more good quality hypotheses tended to ask more recommended questions. This positive association was primarily derived from an increase in non-CSA hypotheses built. The ability to build non-CSA hypotheses was also related to the average number of recommended questions presented by participants. These are encouraging findings and in line with our overall expectations, although not based on a priori hypotheses.

## Perceived reliability of the avatar depends on who is asking the question

On average, interviewers tended towards perceiving the child avatars as better rather than worse witnesses. Compared with conditions without feedback, in conditions with feedback, participants perceived Avatars to be more reliable and less suggestible and believed that the avatars showed fewer contradictions. These differences were obvious in the second half of the interviews. Unsurprisingly, the proportion of recommended questions and the number of

correct details were positively associated with perceived reliability and negatively associated with perceived suggestibility and contradictoriness. The pattern was the opposite for the number of not recommended questions and the number of wrong details. The above results suggest that interviewers were able to have a fairly accurate appraisal of the interview process. To elaborate, based on the predefined algorithms, if the interviewer asked a broad-invitation question (e.g., "Tell me everything you still remember"), the Avatar would respond to this question by retrieving narrative details from their "memory". Then, if the interviewer would have continued to ask a facilitator question (e.g., "And then?") and encouraged the Avatar to keep talking based on what they mentioned before, the Avatar would continue to provide more correct details in their responses. Open-ended questions made the avatars draw information from "recall memory" rather than just selecting "yes" or "no" in response to close-ended questions, which could have introduced false details contradicting other information [15,17,84]. Hence, the perception that the avatars were less reliable and more susceptible when the interviewers employed a larger proportion of not recommended questions is to be expected, and our results confirm that it is noticeable to the participants.

As succinctly summarized by Denne et al. [85], the way in which we question the children about their memories of abuse is a strong predictor of their ability to communicate, and further forms the foundation of their reliability. Our study provides direct evidence for this proposition. On the one hand, it is encouraging that interviewers could form somewhat accurate reliability judgments. On the other hand, the current study offers no insight into whether interviewers are cognizant of the fact that the reliability of the child is linked to their questioning style. In hindsight, it would also be informative to ask interviewers to rate their own interview performance and examine the correlations between self-appraisal and the perceptions of the child avatar.

## Differences between the current and previous studies in the number of questions used

The total number of questions asked by participants in the current study was, on average, 25, which is fewer compared to those asked by European participants, on average 40 [30]. Around 16 questions were asked on average by Japanese participants [27]. Haginoya et al. [27] argued that asking fewer questions of any type could be a strategy Japanese participants adopted to avoid making mistakes in public because of their high sensitivity to negative feedback. However, this might not be the case for the current study. There were no significant differences in the total number of questions when comparing the feedback group to the hypothesis-testing group and control group. Differences in the total number of questions asked were only observed when comparing the combined interventions group to the other three groups, where the combined interventions group asked the fewest questions. A possible explanation is that dual tasks are cognitively more demanding than a single task: When making a decision on what to ask, the combined interventions group must ensure they tested hypotheses with open questions while other groups can go straight forward to the paths where they could keep testing hypotheses with no restriction of question types (hypothesis-testing group), or solely focusing on using more open questions and fewer close-ended questions as reminded by feedback (feedback group), or keep asking whatever they believed can help to find out the truth (control group). The fewer total number of questions asked in the current study compared to previous studies can be explained by the use of fewer non-recommended questions. Participants, in general, used a similar amount of recommended and non-recommended types of questions. While the number of recommended questions in the current study ($M = 12.11$, $SD = 6.93$) is similar to those observed ($M = 13.52$, $SD = 7.96$) in earlier studies (e.g., see [30]), the number

of non recommended questions ($M = 12.38$, $SD = 8.09$) was half of those in an earlier study ($M = 26.39$, $SD = 15.66$). Interestingly, Zeng et al. [86] found that Chinese police officers used a large proportion of open questions (68.9%) when interviewing suspects: The use of open questions differed across three phases, 83.7% during the opening phase, 67.2 during the information-gathering phase, and 43.4% during the closing phase. However, the participants in the current study were university students which would suggest that a general interaction style specific to the Chinese may underlie the finding.

## Limitations and future directions

There are a few more limitations in addition to the one just mentioned above. First, the implementation of the hypothesis-testing procedure was not optimal in the current study. Many of the hypotheses were unfortunately not analytical. Second, given that hypothesis-testing is a cognitively demanding task, it could take the trainees more than two practice rounds to obtain an adequate level of skill. Therefore, the training of hypothesis-testing itself may require further investment to be effective.

The current study lacked sensitivity to elements other than question type when measuring interview quality. The exact questions the interviewers asked were not recorded, so we did not know whether there was a difference in the quality of questions of those in the combined interventions group compared to those who received a single intervention alone. Thus, we recommend that future studies look specifically into the question being asked as well as the self-reported strategy in managing two tasks to gain a better understanding of the execution of hypothesis-testing via interviewer performance, that is, whether the question itself allows testing of hypothesis, as well as of how interviewers, even though with improved questioning skills and good quality information elicited from Avatars, landed on so few completely correct conclusions.

## Conclusions

Taken together, the current study provides further evidence that Avatar Training with feedback is an effective approach for CSA interview training across cultural contexts. Feedback significantly increases the usage of open-ended questions and decreases the usage of close-ended questions, which in turn improves the accuracy of information gathered from the child avatars. Although we fail to observe any positive effects of hypothesis-testing in improving interviewing quality, we caution against the interpretation that hypothesis-testing lacks efficacy as interview quality was only measured in terms of question types. Moreover, we provided direct evidence that children's perceived reliability depends on interviewers' questioning style and encouraged researchers to further explore the interviewer's perceptions and judgments about their interview process.

## Author Contributions

**Conceptualization:** Yiwen Zhang, Shumpei Haginoya, Pekka Olavi Santtila.

**Data curation:** Yiwen Zhang, Siyu Li, Yikang Zhang.

**Formal analysis:** Siyu Li, Yikang Zhang.

**Funding acquisition:** Pekka Olavi Santtila.

**Investigation:** Yiwen Zhang, Siyu Li.

**Methodology:** Yiwen Zhang, Shumpei Haginoya, Pekka Olavi Santtila.

**Project administration:** Yiwen Zhang, Pekka Olavi Santtila.

**Supervision:** Pekka Olavi Santtila.

**Validation:** Siyu Li, Yikang Zhang.

**Visualization:** Siyu Li.

**Writing – original draft:** Yiwen Zhang, Siyu Li, Yikang Zhang, Pekka Olavi Santtila.

**Writing – review & editing:** Yiwen Zhang, Siyu Li, Yikang Zhang, Pekka Olavi Santtila.

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
