## [Decision Letter · Decision Letter 0]

14 Feb 2023

PONE-D-22-24966Effects of combining feedback and hypothesis-testing on the quality of simulated CSA interviews with Avatars among Chinese university studentsPLOS ONE

Dear Dr. Zhang, By my own reading, there’s still the need to detail and better describe Methods (mainly stats parameters & eligibility criteria). Please also work on conciseness, soundness (control of confounding factors) and readability (not that extensive or sentences without updated references).

Also, I’d highly suggest the authors to (i) double-check English, (i) double-check references accordingly to the Journal’s requirements & avoid the use of abbreviations without the first spelling (e.g. the Title) etc. 

Thank you for submitting your manuscript to PLOS ONE. After careful consideration, we feel that it has merit but does not fully meet PLOS ONE’s publication criteria as it currently stands. Therefore, we invite you to submit a revised version of the manuscript that addresses the points raised during the review process.

We look forward to receiving your revised manuscript.

Kind regards,

Thiago Fernandes, PhD

Academic Editor

PLOS ONE

Journal Requirements:

Reviewers' comments:

Reviewer's Responses to Questions

**Comments to the Author**

1. Is the manuscript technically sound, and do the data support the conclusions?

Reviewer #1: Yes

Reviewer #2: Yes

2. Has the statistical analysis been performed appropriately and rigorously? 

Reviewer #1: Yes

Reviewer #2: Yes

3. Have the authors made all data underlying the findings in their manuscript fully available?

Reviewer #1: Yes

Reviewer #2: Yes

4. Is the manuscript presented in an intelligible fashion and written in standard English?

Reviewer #1: Yes

Reviewer #2: Yes

5. Review Comments to the Author

Reviewer #1: - Clear introduction, detailing the relationships that were studied. I would suggest to search in more detail for more recent papers (eg. from 2018 onwards). Well explained hypothesis.

- Well defined population and study design.

- Well defined measures. The data analysis section is also well explained. Statistical analyses also well explained.

- Clear results section. Regarding the discussion part, I would suggest to search in more detail for more recent papers (eg. from 2018 onwards).

- Limitations and Future Research field well explained. Concise conclusion. In general, a very interesting study/article.

Reviewer #2: The authors report a study testing the effect of feedback and hypothesis testing on child sexual assault interview quality. This paper has several strengths that lead me to give it serious consideration for publication in PLOS One. Specifically, I commend the authors for their open science practices (pre-registration and code/data sharing). In addition, this work addresses a very important topic with a rigorous methodology, analysis, and reporting. The main findings support that CSA interview training is effective in China as well (at least so far as college students go) although the effects of the interventions were mixed between being confirmatory/replicating past results (feedback) and non-existent in this study (hypothesis testing) – in fact hypothesis testing by itself may be counter-productive. I hope my reflection on the manuscript are useful to the editor and the authors of this manuscript.

Abstract:

This is clearly written – no comments to give.

Introduction:

Well written.

Results:

Unless I missed it, it isn’t clear to me where you report the analysis for H6 (where you look for improvement in interviewing quality mediating the relationship between HT and interview quality variables. If you did include it, please make it a little clearer for readers. If you did not include a mediation analysis because there was a lack of effect of HT, please explicitly provide this language in the paper for readers.

Were there any corrections for multiple testing made?

Please consider adding p-values in the tables as well (I recognize that only statistically significant findings were put there but it is helpful to put the rest of the findings in proper reference [CI with p-value and effect size]).

I would be interested in seeing a table of the descriptive statistics across the five interviews that you briefly talk about in the respective results section. Here you only give means for the first interview.

Discussion:

It might be helpful for readers if you explicitly talk about each of the hypotheses that you pose in the introduction in a structured fashion.

Very minor, but there are some minor typos throughout the paper that do not detract from the message, however, I imagine that you would all prefer them not to be there. Just one example for the in-text citation:

“The investigative interview is a complex practical skill that places high demands on interviewers (Roberts et ak., 2011). “

6. PLOS authors have the option to publish the peer review history of their article (what does this mean?). If published, this will include your full peer review and any attached files.

Reviewer #1: No

Reviewer #2: No

---

## [Author Response · Author response to Decision Letter 0]

23 Mar 2023

Editor’s comments:

By my own reading, there’s still the need to detail and better describe Methods (mainly stats parameters & eligibility criteria). Please also work on conciseness, soundness (control of confounding factors) and readability (not that extensive or sentences without updated references).

“Many thanks for all your feedback! We have incorporated your suggestions into the manuscript. More recent references have been added. The methods part has also been updated accordingly. The current experimental study leaves very limited room for major confoundings. Full disclosure has been made in terms of previous interview training and experience or having children etc, regarding the random allocation of the participants into the experimental groups.”

Also, I’d highly suggest the authors to (i) double-check English, (i) double-check references accordingly to the Journal’s requirements & avoid the use of abbreviations without the first spelling (e.g. the Title) etc. 

“Double-checking of English, references, and the use of abbreviations is completed.We also double-checked the results section, and made some minor changes. We would like to fully disclose them below:

(1)Page 18 Line 5: t(39.97) = 3.82 revised to t(39.97) = 3.83

(2)Page 24 Line 7: 95% CI [1.08, 1.96] revised to 95% CI [1.08, 1.69]

(3)Page 18 Lines 8-20: Originally, all Standard Deviation (SD) were calculated by using the mean number of hypotheses for each participant among five interviews. We now calculated them by using the raw data in the long format directly.

(4)After using Tukey method to adjust multiple pairwise comparisons, some significant pairwise comparisons became non-significant. We revised both tables and texts accordingly. In the last line of Page 19 to the first line of Page 20, apart from in the 5th interview, we also found that compared to non-HT condition, participants in the HT condition present fewer recommended questions during the 4th interview.

(5)Due to the statistical test of resampling method in function MultRM(), we updated p-value for three-way mixed multivariate analysis of variance (MANOVA) (Please find Page 19 Lines 1-3; Page 20 Lines 22-24; Page 22 Lines 23-25 and Page 23 Line 1).

All revised results have been redlined in the track-change version and did not impact on the main findings of this manuscript.”

Reviewers' comments: 

Reviewer #1: 

- Clear introduction, detailing the relationships that were studied. I would suggest to search in more detail for more recent papers (eg. from 2018 onwards). Well explained hypothesis.

“Thank you for all your help! Changes have been incorporated according to your suggestions. References and citations have been updated with more recent papers.”

- Well defined population and study design

- Well defined measures. The data analysis section is also well explained. Statistical analyses also well explained.

- Clear results section. Regarding the discussion part, I would suggest to search in more detail for more recent papers (eg. from 2018 onwards).

“References and citations have been updated with more recent papers.”

- Limitations and Future Research field well explained. Concise conclusion. In general, a very interesting study/article.

Reviewer #2: 

The authors report a study testing the effect of feedback and hypothesis testing on child sexual assault interview quality. This paper has several strengths that lead me to give it serious consideration for publication in PLOS One. Specifically, I commend the authors for their open science practices (pre-registration and code/data sharing). In addition, this work addresses a very important topic with a rigorous methodology, analysis, and reporting. The main findings support that CSA interview training is effective in China as well (at least so far as college students go) although the effects of the interventions were mixed between being confirmatory/replicating past results (feedback) and non-existent in this study (hypothesis testing) – in fact hypothesis testing by itself may be counter-productive. I hope my reflection on the manuscript are useful to the editor and the authors of this manuscript.

Abstract: This is clearly written – no comments to give.

Introduction: Well written.

Results:

Unless I missed it, it isn’t clear to me where you report the analysis for H6 (where you look for improvement in interviewing quality mediating the relationship between HT and interview quality variables. If you did include it, please make it a little clearer for readers. If you did not include a mediation analysis because there was a lack of effect of HT, please explicitly provide this language in the paper for readers.

“It has now been made clearer in the revised manuscript regarding the analysis for H6 not being performed due to a lack of the effect of HT.”

Were there any corrections for multiple testing made?

“Thank you so much for pointing this out. We have revised the results section (i.e., Table 6, Table 7, and Table 8) by using the Tukey method to adjust for multiple comparisons. After adjustment, some pairwise comparisons became non-significant. We removed them from Tables 6-8 and updated texts accordingly.”

Please consider adding p-values in the tables as well (I recognize that only statistically significant findings were put there but it is helpful to put the rest of the findings in proper reference [CI with p-value and effect size]).

“Thank you so much for your comment. We have added p-values to Tables 3-8. Additionally, all non-significant results of pairwise comparisons are presented at Open Science Framework (OSF): https://osf.io/7ytnh.”

I would be interested in seeing a table of the descriptive statistics across the five interviews that you briefly talk about in the respective results section. Here you only give means for the first interview.

“Thank you so much for your suggestion. We agreed that it is necessary to present the results of descriptive statistics for the overall five interviews. Given that the correlation tables were presented separately, we added some sentences to the descriptive statistics section to present Means (M) and Standard Deviations (SD) for the overall five interviews in the text rather than adding them to the correlation tables directly (From Page 15: Lines 21-28 to Page 16: Lines 1-2). Please see added contents below:

‘On average, the number of recommended questions used by participants in each interview (M = 12.11, SD = 8.83) was roughly the same as the number of non-recommended questions (M = 12.38, SD = 9.57): the proportion of recommended questions was 50.29% (27.89%). Participants on average elicited 2.71 relevant details (SD = 2.39), 2.77 neutral details (SD = 2.51), and 3.04 wrong details (SD = 3.04). Additionally, concerning all five interviews, participants assessed avatars’ reliability with the highest score (M = 66.86, SD = 18.25) compared to assessed avatars’ contradictory (M = 26,21, SD = 28.88) and suggestibility (M = 43.00, SD = 23.56). These results did not show a clear numeric superiority of recommended questions among five interviews overall, and our participants failed to elicit a greater number of relevant details compared to neutral and wrong details from avatars.’”

Discussion:

It might be helpful for readers if you explicitly talk about each of the hypotheses that you pose in the introduction in a structured fashion.

“Changes have been made to explicitly state whether the hypotheses posed in the introduction were supported in a structured fashion.”

Very minor, but there are some minor typos throughout the paper that do not detract from the message, however, I imagine that you would all prefer them not to be there. Just one example for the in-text citation:

“The investigative interview is a complex practical skill that places high demands on interviewers (Roberts et ak., 2011). “

“Corrections have been made.”

---

## [Decision Letter · Decision Letter 1]

11 Apr 2023

PONE-D-22-24966R1Effects of combining feedback and hypothesis-testing on the quality of simulated CSA interviews with Avatars among Chinese university studentsPLOS ONE

Dear Dr. Zhang,

Thank you for submitting your manuscript to PLOS ONE. After careful consideration, we feel that it has merit but does not fully meet PLOS ONE’s publication criteria as it currently stands. Therefore, we invite you to submit a revised version of the manuscript that addresses the points raised during the review process. Thank you for your valuable study. The remaining concerns are interesting and important, and also I think won't take too long. As long as the authors are willing to review, please submit a detailed list of changes for each point raised when you submit the revised manuscript. Please also highlight where the text has been changed in the resubmitted file - this will help to streamline the reviewing process and minimise any delays. Another suggestion is: avoid the use of abbreviation in Title. Remember, whenever possible, to explain your abbreviations and spell them first. Looking forward for your updated version.

We look forward to receiving your revised manuscript.

Kind regards,

Thiago P. Fernandes, PhD

Academic Editor

PLOS ONE

Journal Requirements:

Reviewers' comments:

Reviewer's Responses to Questions

**Comments to the Author**

1. If the authors have adequately addressed your comments raised in a previous round of review and you feel that this manuscript is now acceptable for publication, you may indicate that here to bypass the “Comments to the Author” section, enter your conflict of interest statement in the “Confidential to Editor” section, and submit your "Accept" recommendation.

Reviewer #2: All comments have been addressed

Reviewer #3: (No Response)

2. Is the manuscript technically sound, and do the data support the conclusions?

Reviewer #2: Yes

Reviewer #3: Yes

3. Has the statistical analysis been performed appropriately and rigorously? 

Reviewer #2: Yes

Reviewer #3: Yes

4. Have the authors made all data underlying the findings in their manuscript fully available?

Reviewer #2: Yes

Reviewer #3: Yes

5. Is the manuscript presented in an intelligible fashion and written in standard English?

Reviewer #2: Yes

Reviewer #3: Yes

6. Review Comments to the Author

Reviewer #2: Great job with this paper. Thank you for addressing the comments that were left for you. Best of luck.

Reviewer #3: This study set out to test the effects of different conditions on interview quality regarding child abuse.

The manuscript is well-written and well-structured.

It is easy to understand the questions and hypotheses, as their presentation in the Intro and Discussion was very clear and organised.

The Ideas expressed are interesting and appear to add to the current Literature.

There are only some minor corrections necessary (see below).

Some corrections:

In the abstract: “Eight-one Chinese university students” -> “Eighty-one…”

In the participant section: “18 years old or higher” -> “18 years old or older”

In the example given in Methods: “Nicholas seems to be at ease is during dance lessons, where his mother pushed him to start at the age of 4” -> “…lessons, which his mother…”, AND “…the new teacher Richard arrived at dance school…” -> “… arrived at the dance school…”

In the results section: “To look at the effects of the HT and Feedback interventions, respectively, in each number of interview…” -> “…respectively, in in each group of interviews ….”, if this is what is meant. Otherwise -> “in each interview”

In Conclusions: First sentence says “… a statistically positive correlation with…” -> “…statistically significant positive correlation…”

In Discussion: “In the following sections, we briefly discussed the findings, their implications, and the limitations of the current research.” -> “…we briefly discuss…”

ALSO “On one hand, it is encouraging that interviewers could form somewhat accurate reliability judgments.” -> “On the one hand…”

All in all, the study addresses an important issue by implementing current and modern methodologies (e.g., training with computer-simulated Avatars with the addition of hypothesis-testing intervention), it makes useful remarks for future research based on the obtained results, and provides an excellent description of the (and, therefore, an easy-to-repeat) statistical analysis model (which can, potentially, be useful in future research).

7. PLOS authors have the option to publish the peer review history of their article (what does this mean?). If published, this will include your full peer review and any attached files.

Reviewer #2: No

Reviewer #3: No

---

## [Author Response · Author response to Decision Letter 1]

13 Apr 2023

Round 1 

Editor’s comments:

By my own reading, there’s still the need to detail and better describe Methods (mainly stats parameters & eligibility criteria). Please also work on conciseness, soundness (control of confounding factors) and readability (not that extensive or sentences without updated references).

“Many thanks for all your feedback! We have incorporated your suggestions into the manuscript. More recent references have been added. The methods part has also been updated accordingly. The current experimental study leaves very limited room for major confoundings. Full disclosure has been made in terms of previous interview training and experience or having children etc, regarding the random allocation of the participants into the experimental groups.”

Also, I’d highly suggest the authors to (i) double-check English, (i) double-check references accordingly to the Journal’s requirements & avoid the use of abbreviations without the first spelling (e.g. the Title) etc. 

“Double-checking of English, references, and the use of abbreviations is completed.We also double-checked the results section, and made some minor changes. We would like to fully disclose them below:

(1)Page 18 Line 5: t(39.97) = 3.82 revised to t(39.97) = 3.83

(2)Page 24 Line 7: 95% CI [1.08, 1.96] revised to 95% CI [1.08, 1.69]

(3)Page 18 Lines 8-20: Originally, all Standard Deviation (SD) were calculated by using the mean number of hypotheses for each participant among five interviews. We now calculated them by using the raw data in the long format directly.

(4)After using Tukey method to adjust multiple pairwise comparisons, some significant pairwise comparisons became non-significant. We revised both tables and texts accordingly. In the last line of Page 19 to the first line of Page 20, apart from in the 5th interview, we also found that compared to non-HT condition, participants in the HT condition present fewer recommended questions during the 4th interview.

(5)Due to the statistical test of resampling method in function MultRM(), we updated p-value for three-way mixed multivariate analysis of variance (MANOVA) (Please find Page 19 Lines 1-3; Page 20 Lines 22-24; Page 22 Lines 23-25 and Page 23 Line 1).

All revised results have been redlined in the track-change version and did not impact on the main findings of this manuscript.”

Reviewers' comments: 

Reviewer #1: 

- Clear introduction, detailing the relationships that were studied. I would suggest to search in more detail for more recent papers (eg. from 2018 onwards). Well explained hypothesis.

“Thank you for all your help! Changes have been incorporated according to your suggestions. References and citations have been updated with more recent papers.”

- Well defined population and study design

- Well defined measures. The data analysis section is also well explained. Statistical analyses also well explained.

- Clear results section. Regarding the discussion part, I would suggest to search in more detail for more recent papers (eg. from 2018 onwards).

“References and citations have been updated with more recent papers.”

- Limitations and Future Research field well explained. Concise conclusion. In general, a very interesting study/article.

Reviewer #2: 

The authors report a study testing the effect of feedback and hypothesis testing on child sexual assault interview quality. This paper has several strengths that lead me to give it serious consideration for publication in PLOS One. Specifically, I commend the authors for their open science practices (pre-registration and code/data sharing). In addition, this work addresses a very important topic with a rigorous methodology, analysis, and reporting. The main findings support that CSA interview training is effective in China as well (at least so far as college students go) although the effects of the interventions were mixed between being confirmatory/replicating past results (feedback) and non-existent in this study (hypothesis testing) – in fact hypothesis testing by itself may be counter-productive. I hope my reflection on the manuscript are useful to the editor and the authors of this manuscript.

Abstract: This is clearly written – no comments to give.

Introduction: Well written.

Results:

Unless I missed it, it isn’t clear to me where you report the analysis for H6 (where you look for improvement in interviewing quality mediating the relationship between HT and interview quality variables. If you did include it, please make it a little clearer for readers. If you did not include a mediation analysis because there was a lack of effect of HT, please explicitly provide this language in the paper for readers.

“It has now been made clearer in the revised manuscript regarding the analysis for H6 not being performed due to a lack of the effect of HT.”

Were there any corrections for multiple testing made?

“Thank you so much for pointing this out. We have revised the results section (i.e., Table 6, Table 7, and Table 8) by using the Tukey method to adjust for multiple comparisons. After adjustment, some pairwise comparisons became non-significant. We removed them from Tables 6-8 and updated texts accordingly.”

Please consider adding p-values in the tables as well (I recognize that only statistically significant findings were put there but it is helpful to put the rest of the findings in proper reference [CI with p-value and effect size]).

“Thank you so much for your comment. We have added p-values to Tables 3-8. Additionally, all non-significant results of pairwise comparisons are presented at Open Science Framework (OSF): https://osf.io/7ytnh.”

I would be interested in seeing a table of the descriptive statistics across the five interviews that you briefly talk about in the respective results section. Here you only give means for the first interview.

“Thank you so much for your suggestion. We agreed that it is necessary to present the results of descriptive statistics for the overall five interviews. Given that the correlation tables were presented separately, we added some sentences to the descriptive statistics section to present Means (M) and Standard Deviations (SD) for the overall five interviews in the text rather than adding them to the correlation tables directly (From Page 15: Lines 21-28 to Page 16: Lines 1-2). Please see added contents below:

‘On average, the number of recommended questions used by participants in each interview (M = 12.11, SD = 8.83) was roughly the same as the number of non-recommended questions (M = 12.38, SD = 9.57): the proportion of recommended questions was 50.29% (27.89%). Participants on average elicited 2.71 relevant details (SD = 2.39), 2.77 neutral details (SD = 2.51), and 3.04 wrong details (SD = 3.04). Additionally, concerning all five interviews, participants assessed avatars’ reliability with the highest score (M = 66.86, SD = 18.25) compared to assessed avatars’ contradictory (M = 26,21, SD = 28.88) and suggestibility (M = 43.00, SD = 23.56). These results did not show a clear numeric superiority of recommended questions among five interviews overall, and our participants failed to elicit a greater number of relevant details compared to neutral and wrong details from avatars.’”

Discussion:

It might be helpful for readers if you explicitly talk about each of the hypotheses that you pose in the introduction in a structured fashion.

“Changes have been made to explicitly state whether the hypotheses posed in the introduction were supported in a structured fashion.”

Very minor, but there are some minor typos throughout the paper that do not detract from the message, however, I imagine that you would all prefer them not to be there. Just one example for the in-text citation:

“The investigative interview is a complex practical skill that places high demands on interviewers (Roberts et ak., 2011). “

“Corrections have been made.”

Round 2 

Editor’s comments:

Thank you for your valuable study. The remaining concerns are interesting and important, and also I think won't take too long. As long as the authors are willing to review, please submit a detailed list of changes for each point raised when you submit the revised manuscript. Please also highlight where the text has been changed in the resubmitted file - this will help to streamline the reviewing process and minimise any delays.

Another suggestion is: avoid the use of abbreviation in Title. 

Remember, whenever possible, to explain your abbreviations and spell them first.

“Thank you for your feedback! Changes have been highlighted and abbreviation in Title has been adjusted.”

Reviewer #3:

This study set out to test the effects of different conditions on interview quality regarding child abuse.

The manuscript is well-written and well-structured.

It is easy to understand the questions and hypotheses, as their presentation in the Intro and Discussion was very clear and organised.

The Ideas expressed are interesting and appear to add to the current Literature.

There are only some minor corrections necessary (see below).

Some corrections:

In the abstract: “Eight-one Chinese university students” -> “Eighty-one…”

In the participant section: “18 years old or higher” -> “18 years old or older”

In the example given in Methods: “Nicholas seems to be at ease is during dance lessons, where his mother pushed him to start at the age of 4” -> “…lessons, which his mother…”, AND “…the new teacher Richard arrived at dance school…” -> “… arrived at the dance school…”

In the results section: “To look at the effects of the HT and Feedback interventions, respectively, in each number of interview…” -> “…respectively, in in each group of interviews ….”, if this is what is meant. Otherwise -> “in each interview”

In Conclusions: First sentence says “… a statistically positive correlation with…” -> “…statistically significant positive correlation…”

In Discussion: “In the following sections, we briefly discussed the findings, their implications, and the limitations of the current research.” -> “…we briefly discuss…”

ALSO “On one hand, it is encouraging that interviewers could form somewhat accurate reliability judgments.” -> “On the one hand…”

All in all, the study addresses an important issue by implementing current and modern methodologies (e.g., training with computer-simulated Avatars with the addition of hypothesis-testing intervention), it makes useful remarks for future research based on the obtained results, and provides an excellent description of the (and, therefore, an easy-to-repeat) statistical analysis model (which can, potentially, be useful in future research).

“Thank you for your feedback! Corrections have been made.”

---

## [Editor Report · Decision Letter 2]

16 Apr 2023

Effects of combining feedback and hypothesis-testing on the quality of simulated child sexual abuse interviews with Avatars among Chinese university students

PONE-D-22-24966R2

Dear Dr. Zhang,

We’re pleased to inform you that your manuscript has been judged scientifically suitable for publication and will be formally accepted for publication once it meets all outstanding technical requirements.

Kind regards,

Thiago P. Fernandes, PhD

Academic Editor

PLOS ONE
---

## [Editor Report · Acceptance letter]

19 Apr 2023

PONE-D-22-24966R2 

Effects of combining feedback and hypothesis-testing on the quality of simulated child sexual abuse interviews with Avatars among Chinese university students 

Dear Dr. Zhang:

I'm pleased to inform you that your manuscript has been deemed suitable for publication in PLOS ONE. Congratulations! Your manuscript is now with our production department. 

Kind regards, 

on behalf of

Dr. Thiago P. Fernandes 

Academic Editor

PLOS ONE